# Adapting to Function Difficulty and Growth Conditions in Private Optimization

**Hilal Asi**[*]    **Daniel Levy**[*]    **John C. Duchi**
{asi,danilevy,jduchi}@stanford.edu

## Abstract

We develop algorithms for private stochastic convex optimization that adapt to the hardness of the specific function we wish to optimize. While previous work provide worst-case bounds for arbitrary convex functions, it is often the case that the function at hand belongs to a smaller class that enjoys faster rates. Concretely, we show that for functions exhibiting $\kappa$-growth around the optimum, i.e., $f(x) \geq f(x^\star) + \lambda \kappa^{-1} \|x - x^\star\|_2^\kappa$ for $\kappa > 1$, our algorithms improve upon the standard $\sqrt{d}/n\varepsilon$ privacy rate to the faster $(\sqrt{d}/n\varepsilon)^{\frac{\kappa}{\kappa-1}}$. Crucially, they achieve these rates without knowledge of the growth constant $\kappa$ of the function. Our algorithms build upon the inverse sensitivity mechanism, which adapts to instance difficulty [2], and recent localization techniques in private optimization [25]. We complement our algorithms with matching lower bounds for these function classes and demonstrate that our adaptive algorithm is *simultaneously* (minimax) optimal over all $\kappa \geq 1 + c$ whenever $c = \Theta(1)$.

## 1 Introduction

Stochastic convex optimization (SCO) is a central problem in machine learning and statistics, where for a sample space $\mathbb{S}$, parameter space $\mathcal{X} \subset \mathbb{R}^d$, and a collection of convex losses $\{F(\cdot; s) : s \in \mathbb{S}\}$, one wishes to solve

$$\operatorname*{minimize}_{x \in \mathcal{X}} f(x) \coloneqq \mathbb{E}_{S \sim P}[F(x; S)] = \int_{\mathbb{S}} F(x; s) \mathrm{d}P(s) \tag{1}$$

using an observed dataset $\mathcal{S} = S_1^n \overset{\text{iid}}{\sim} P$. While as formulated, the problem is by now fairly well-understood [12, 38, 29, 10, 37], it is becoming clear that, because of considerations beyond pure statistical accuracy—memory or communication costs [45, 26, 13], fairness [23, 28], personalization or distributed learning [35]—problem (1) is simply insufficient to address modern learning problems. To that end, researchers have revisited SCO under the additional constraint that the solution preserves the privacy of the provided sample [22, 21, 1, 16, 19]. A waypoint is Bassily et al. [7], who provide a private method with optimal convergence rates for the related empirical risk minimization problem, with recent papers focus on SCO providing (worst-case) optimal rates in various settings: smooth convex functions [8, 25], non-smooth functions [9], non-Euclidean geometry [5, 4] and under more stringent privacy constraints [34].

Yet these works ground their analyses in worst-case scenarios and provide guarantees for the *hardest* instance of the class of problems they consider. Conversely, they argue that their algorithms are optimal in a minimax sense: for any algorithm, there exists a hard instance on which the error achieved by the algorithm is equal to the upper bound. While valuable, these results are pessimistic—the exhibited hard instances are typically pathological—and fail to reflect achievable performance.

---

[*]Equal contribution. Author order determined by coin toss.

35th Conference on Neural Information Processing Systems (NeurIPS 2021).

In this work, we consider the problem of adaptivity when solving (1) under privacy constraints. Importantly, we wish to provide private algorithms that *adapt* to the hardness of the objective $f$. A loss function $f$ may belong to multiple problem classes, each exhibiting different achievable rates, so a natural desideratum is to attain the error rate of the easiest sub-class. As a simple vignette, if one gets an arbitrary 1-Lipschitz convex loss function $f$, the worst-case guarantee of any $\varepsilon$-DP algorithm is $\Theta(1/\sqrt{n} + d/(n\varepsilon))$. However, if one learns that $f$ exhibits some growth property—say $f$ is 1-strongly convex—the regret guarantee improves to the faster $\Theta(1/n + (d/(n\varepsilon))^2)$ rate with the appropriate algorithm. It is thus important to provide algorithms that achieves the rates of the "easiest" class to which the function belongs [32, 46, 18].

To that end, consider the nested classes of functions $\mathcal{F}^\kappa$ for $\kappa \in [1, \infty]$ such that, if $f \in \mathcal{F}^\kappa$ then there exists $\lambda > 0$ such that for all $x \in \mathcal{X}$,

$$f(x) - \inf_{x' \in \mathcal{X}} f(x') \geq \frac{\lambda}{\kappa} \|x - x^\star\|_2^\kappa.$$

For example, strong convexity implies growth with parameter $\kappa = 2$. This growth assumption closely relates to uniform convexity [32] and the Polyak-Kurdyka-Łojasiewicz inequality [11], and we make these connections precise in Section 2. Intuitively, smaller $\kappa$ makes the function much easier to optimize: the error around the optimal point grows quickly. Objectives with growth are widespread in machine learning applications: among others, the $\ell_1$-regularized hinge loss exhibits sharp growth (i.e. $\kappa = 1$) while $\ell_1$- or $\ell_\infty$-constrained $\kappa$-norm regression —i.e. $s = (a, b) \in \mathbb{R}^d \times \mathbb{R}$ and $F(x; s) = |b - \langle a, x \rangle|^\kappa$—has $\kappa$-growth for any $\kappa$ integer greater than 2 [43]. In this work, we provide private adaptive algorithms that adapt to the *actual* growth of the function at hand.

We begin our analysis by examining Asi and Duchi's inverse sensitivity mechanism [2] on ERM as a motivation. While not a practical algorithm, it achieves instance-optimal rates for any one-dimensional function under mild assumptions, quantifying the best bound one could hope to achieve with an adaptive algorithm, and showing (in principle) that adaptive private algorithms can exist. We first show that for any function with $\kappa$-growth, the inverse sensitivity mechanism achieves privacy cost $(d/(n\varepsilon))^{\kappa/(\kappa-1)}$; importantly, *without knowledge of the function class $\mathcal{F}^\kappa$, that $f$ belongs to*. This constitutes grounding and motivation for our work in three ways: (i) it validates our choice of sub-classes $\mathcal{F}^\kappa$ as the privacy rate is effectively controlled by the value of $\kappa$, (ii) it exhibits the rate we wish to achieve with efficient algorithms on $\mathcal{F}^\kappa$ and (iii) it showcases that for easier functions, privacy costs shrink significantly—to illustrate, for $\kappa = 5/4$ the privacy rate becomes $(d/(n\varepsilon))^5$.

We continue our treatment of problem (1) under growth in Section 4 and develop practical algorithms that achieve the rates of the inverse sensitivity mechanism. Moreover, for approximate $(\varepsilon, \delta)$-differential privacy, our algorithms improve the rates, achieving roughly $(\sqrt{d}/(n\varepsilon))^{\kappa/(\kappa-1)}$. Our algorithms hinge on a reduction to SCO: we show that by solving a sequence of increasingly constrained SCO problems, one achieves the right rate whenever the function exhibits growth at the optimum. Importantly, our algorithm only requires a *lower bound* $\underline{\kappa} \leq \kappa$ (where $\kappa$ is the actual growth of $f$).

We provide optimality guarantees for our algorithms in Section 5 and show that both the inverse sensitivity and the efficient algorithms of Section 4 are *simultaneously minimax optimal* over all classes $\mathcal{F}^\kappa$ whenever $\kappa = 1 + \Theta(1)$ and $d = 1$ for $\varepsilon$-DP algorithms. Finally, we prove that in *arbitrary dimension*, for both pure- and approximate-DP constraints, our algorithms are also simultaneously optimal for all classes $\mathcal{F}^\kappa$ with $\kappa \geq 2$.

On the way, we provide results that may be of independent interest to the community. First, we develop optimal algorithms for SCO under *pure* differential privacy constraints, which, to the best of our knowledge, do not exist in the literature. Secondly, our algorithms and analysis provide high-probability bounds on the loss, whereas existing results only provide (weaker) bounds on the expected loss. Finally, we complete the results of Ramdas and Singh [40] on (non-private) optimization lower bounds for functions with $\kappa$-growth by providing information-theoretic lower bounds (in contrast to oracle-based lower bounds that rely on observing only gradient information) and capturing the optimal dependence on all problem parameters (namely $d$, $L$ and $\lambda$).

## 1.1 Related work

Convex optimization is one of the best studied problems in private data analysis [16, 19, 41, 7]. The first papers in this line of work mainly study minimizing the empirical loss, and readily establish that

the (minimax) optimal privacy rates are $d/n\varepsilon$ for pure $\varepsilon$-DP and $\sqrt{d\log(1/\delta)}/n\varepsilon$ for $(\varepsilon,\delta)$-DP [16, 7]. More recently, several works instead consider the harder problem of privately minimizing the population loss [8, 25]. These papers introduce new algorithmic techniques to obtain the worst-case optimal rates of $1/\sqrt{n} + \sqrt{d\log(1/\delta)}/n\varepsilon$ for $(\varepsilon,\delta)$-DP. They also show how to improve this rate to the faster $1/n + d\log(1/\delta)/(n\varepsilon)^2$ in the case of 1-strongly convex functions. Our work subsumes both of these results as they correspond to $\kappa = \infty$ and $\kappa = 2$ respectively. To the best of our knowledge, there has been no work in private optimization that investigates the rates under general $\kappa$-growth assumptions or adaptivity to such conditions.

In contrast, the optimization community has extensively studied growth assumptions [40, 32, 15] and show that on these problems, carefully crafted algorithms improves upon the standard $1/\sqrt{n}$ for convex functions to the faster $(1/\sqrt{n})^{\kappa/(\kappa-1)}$. [32] derives worst-case optimal (in the first-order oracle model) gradient algorithms in the uniformly convex case (i.e. $\kappa \geq 2$) and provides technique to adapt to the growth $\kappa$, while [40], drawing connections between growth conditions and active learning, provides upper and lower bounds in the first-order stochastic oracle model. We complete the results of the latter and provide *information-theoretic* lower bounds that have optimal dependence on $d$, $\lambda$ and $n$—their lower bound only holding for $\lambda$ inversely proportional to $d^{1/2-1/\kappa}$, when $\kappa \geq 2$. Closest to our work is [15] who studies instance-optimality via local minimax complexity [14]. For one-dimensional functions, they develop a bisection-based instance-optimal algorithm and show that on individual functions of the form $t \mapsto \kappa^{-1}|t|^\kappa$, the local minimax rate is $(1/\sqrt{n})^{\kappa/(\kappa-1)}$.

## 2 Preliminaries

We first provide notation that we use throughout this paper, define useful assumptions and present key definitions in convex analysis and differential privacy.

**Notation.**  $n$ typically denotes the sample size and $d$ the dimension. Throughout this work, $x$ refers to the optimization variable, $\mathcal{X} \subset \mathbb{R}^d$ to the constraint set and $s$ to elements ($S$ when random) of the sample space $\mathbb{S}$. We usually denote by $F : \mathcal{X} \times \mathbb{S} \to \mathbb{R}$ the (convex) loss function and for a dataset $\mathcal{S} = (s_1, \ldots, s_n) \subset \mathbb{S}$, we define the empirical and population losses

$$f_{\mathcal{S}}(x) := \frac{1}{n}\sum_{i \leq n} F(x; s_i) \ \text{ and } \ f(x) := \mathbb{E}_{S \sim P}[F(x; S)].$$

We omit the dependence on $P$ as it is often clear from context. We reserve $\varepsilon, \delta \geq 0$ for the privacy parameters of Definition 2.1. We always take gradients with respect to the optimization variable $x$. In the case that $F(\cdot; s)$ is not differentiable at $x$, we override notation and define $\nabla F(x; s) = \operatorname{argmin}_{g \in \partial F(x;s)} \|g\|_2$, where $\partial F(x; s)$ is the subdifferential of $F(\cdot; s)$ at $x$. We use A for (potentially random) mechanism and $S_1^n$ as a shorthand for $(S_1, \ldots, S_n)$. For $p \geq 1$, $\|\cdot\|_p$ is the standard $\ell_p$-norm, $\mathbb{B}_p^d(R)$ is the corresponding $d$-dimensional $p$-ball of radius $R$ and $p^\star$ is the dual of $p$, i.e. such that $1/p^\star + 1/p = 1$. Finally, we define the Hamming distance between datasets $d_{\mathrm{Ham}}(\mathcal{S}, \mathcal{S}') := \inf_{\sigma \in \mathfrak{S}_n} \mathbf{1}\{s_i \neq s'_{\sigma(i)}\}$, where $\mathfrak{S}_n$ is the set of permutations over sets of size $n$.

**Assumptions.**  We first state standard assumptions for solving (1). We assume that $\mathcal{X}$ is a closed, convex domain such that $\mathrm{diam}_2(\mathcal{X}) = \sup_{x,y \in \mathcal{X}} \|x - y\|_2 \leq D < \infty$. Furthermore, we assume that for any $s \in \mathbb{S}$, $F(\cdot; s)$ is convex and $L$-Lipschitz with respect to $\|\cdot\|_2$. Central to our work, we define the following $\kappa$-growth assumption.

**Assumption 1** ($\kappa$-growth). *Let $x^\star = \operatorname{argmin}_{x \in \mathcal{X}} f(x)$. For a loss $F$ and distribution $P$, we say that $(F, P)$ has $(\lambda, \kappa)$ growth for $\kappa \in [1, \infty]$ and $\lambda > 0$, if the population function satisfies*

$$\text{for all } x \in \mathcal{X}, \quad f(x) - f(x^\star) \geq \frac{\lambda}{\kappa}\|x - x^\star\|_2^\kappa.$$

*In the case where $\widehat{P}$ is the empirical distribution on a finite dataset $\mathcal{S}$, we refer to $(\lambda, \kappa)$-growth of $(F, \widehat{P})$ as $\kappa$-growth of the empirical function $f_{\mathcal{S}}$.*

**Uniform convexity and Kurdyka-Łojasiewicz inequality.**  Assumption 1 is closely related to two fundamental notions in convex analysis: uniform convexity and the Kurdyka-Łojasiewicz inequality.

Following [39], we say that $h : \mathcal{Z} \subset \mathbb{R}^d \to \mathbb{R}$ is $(\sigma, \kappa)$-uniformly convex with $\sigma > 0$ and $\kappa \geq 2$ if

$$\text{for all } x, y \in \mathcal{Z}, \quad h(y) \geq h(x) + \langle \nabla h(x), y - x \rangle + \frac{\sigma}{\kappa} \|x - y\|_2^\kappa.$$

This immediately implies that (i) sums (and expectations) preserve uniform convexity (ii) if $f$ is uniformly convex with $\lambda$ and $\kappa$, then it has $(\lambda, \kappa)$-growth. This will be useful when constructing hard instances as it will suffice to consider $(\lambda, \kappa)$-uniformly convex functions which are generally more convenient to manipulate. Finally, we point out that, in the general case that $\kappa \geq 1$, the literature refers to Assumption 1 as the Kurdyka-Łojasiewicz inequality [11] with, in their notation, $\varphi(s) = (\kappa/\lambda)^{1/\kappa} s^{1/\kappa}$. Theorem 5-(ii) in [11] says that, under mild conditions, Assumption 1 implies the following inequality between the error and the gradient norm for all $x \in \mathcal{X}$

$$f(x) - \inf_{x' \in \mathcal{X}} f(x') \leq \frac{e}{\lambda^{\frac{1}{\kappa-1}}} \|\nabla f(x)\|_2^{\frac{\kappa}{\kappa-1}}, \tag{2}$$

This is a key result in our analysis of the inverse sensitivity mechanism of Section 3.

**Differential privacy.** We begin by recalling the definition of $(\varepsilon, \delta)$-differential privacy.

**Definition 2.1** ([22, 21]). *A randomized algorithm* $\mathsf{A}$ *is* $(\varepsilon, \delta)$-*differentially private* $((\varepsilon, \delta)$-*DP) if, for all datasets* $\mathcal{S}, \mathcal{S}' \in \mathbb{S}^n$ *that differ in a single data element and for all events* $\mathcal{O}$ *in the output space of* $\mathsf{A}$, *we have*

$$\Pr\left(\mathsf{A}(\mathcal{S}) \in \mathcal{O}\right) \leq e^\epsilon \Pr\left(\mathsf{A}(\mathcal{S}') \in \mathcal{O}\right) + \delta.$$

We use the following standard results in differential privacy.

**Lemma 2.1** (Composition [20, Thm. 3.16]). *If* $\mathsf{A}_1, \ldots, \mathsf{A}_k$ *are randomized algorithms that each is* $(\varepsilon, \delta)$-*DP, then their composition* $(\mathsf{A}_1(\mathcal{S}), \ldots, \mathsf{A}_k(\mathcal{S}))$ *is* $(k\varepsilon, k\delta)$-*DP.*

Next, we consider the Laplace mechanism. We will let $Z \sim \mathsf{Lap}_d(\sigma)$ denote a $d$-dimensional vector $Z \in \mathbb{R}^d$ such that $Z_i \overset{\text{iid}}{\sim} \mathsf{Lap}(\sigma)$ for $1 \leq i \leq d$.

**Lemma 2.2** (Laplace mechanism [20, Thm. 3.6]). *Let* $h : \mathbb{S}^n \to \mathbb{R}^d$ *have* $\ell_1$-*sensitivity* $\Delta$, *that is,* $\sup_{\mathcal{S}, \mathcal{S}' \in \mathbb{S}^n : d_{\text{Ham}}(\mathcal{S}, \mathcal{S}') \leq 1} \|h(\mathcal{S}) - h(\mathcal{S}')\|_1 \leq \Delta$. *Then the Laplace mechanism* $\mathsf{A}(\mathcal{S}) = h(\mathcal{S}) + \mathsf{Lap}_d(\sigma)$ *with* $\sigma = \Delta/\varepsilon$ *is* $\varepsilon$-*DP.*

Finally, we need the Gaussian mechanism for $(\varepsilon, \delta)$-DP.

**Lemma 2.3** (Gaussian mechanism [20, Thm. A.1]). *Let* $h : \mathbb{S}^n \to \mathbb{R}^d$ *have* $\ell_2$-*sensitivity* $\Delta$, *that is,* $\sup_{\mathcal{S}, \mathcal{S}' \in \mathbb{S}^n : d_{\text{Ham}}(\mathcal{S}, \mathcal{S}') \leq 1} \|h(\mathcal{S}) - h(\mathcal{S}')\|_2 \leq \Delta$. *Then the Gaussian mechanism* $\mathsf{A}(\mathcal{S}) = h(\mathcal{S}) + \mathsf{N}(0, \sigma^2 I_d)$ *with* $\sigma = 2\Delta \log(2/\delta)/\varepsilon$ *is* $(\varepsilon, \delta)$-*DP.*

**Inverse sensitivity mechanism.** Our goal is to design private optimization algorithms that adapt to the difficulty of the underlying function. As a reference point, we turn to the inverse sensitivity mechanism of [2] as it enjoys general instance-optimality guarantees. For a given function $h : \mathbb{S}^n \to \mathcal{T} \subset \mathbb{R}^d$ that we wish to estimate privately, define the *inverse sensitivity* at $x \in \mathcal{T}$

$$\mathsf{len}_h(\mathcal{S}; x) = \inf_{\mathcal{S}'} \{d_{\text{Ham}}(\mathcal{S}', \mathcal{S}) : h(\mathcal{S}') = x\}, \tag{3}$$

that is, the inverse sensitivity of a target parameter $y \in \mathcal{T}$ at instance $\mathcal{S}$ is the minimal number of samples one needs to change to reach a new instance $\mathcal{S}'$ such that $h(\mathcal{S}') = y$. Having this quantity, the inverse sensitivity mechanism samples an output from the following probability density

$$\pi_{\mathsf{A}_{\text{inv}}(\mathcal{S})}(x) \propto e^{-\varepsilon \mathsf{len}_h(\mathcal{S}; x)}. \tag{4}$$

The inverse sensitivity mechanism preserves $\varepsilon$-DP and enjoys instance-optimality guarantees in general settings [2]. In contrast to (worst-case) minimax optimality guarantees which measure the performance of the algorithm on the hardest instance, these notions of instance-optimality provide stronger per-instance optimality guarantees.

# 3 Adaptive rates through inverse sensitivity for $\varepsilon$-DP

To understand the achievable rates when privately optimizing functions with growth, we begin our theoretical investigation by examining the inverse sensitivity mechanism in our setting. We show that, for instances that exhibit $\kappa$-growth of the empirical function, the inverse sensitivity mechanism privately solves ERM with excess loss roughly $(d/n\varepsilon)^{\frac{\kappa}{\kappa-1}}$.

In our setting, we use a gradient-based approximation of the inverse sensitivity mechanism to simplify the analysis, while attaining similar rates. Following [3] with our function of interest $h(\mathcal{S}) := \operatorname{argmin}_{x \in \mathcal{X}} f_{\mathcal{S}}(x)$, we can lower bound the inverse sensitivity $\operatorname{len}_h(\mathcal{S}; x) \geq n\|\nabla f_{\mathcal{S}}(x)\|_2/2L$ under natural assumptions. We define a $\rho$-smoothed version of this quantity which is more suitable to continuous domains

$$G_{\mathcal{S}}^\rho(x) = \inf_{y \in \mathcal{X} : \|y-x\|_2 \leq \rho} \|\nabla f_{\mathcal{S}}(y)\|_2,$$

and define the $\rho$-smooth gradient-based inverse sensitivity mechanism

$$\pi_{\mathsf{A}_{\mathrm{gr-inv}}(\mathcal{S})}(x) \propto e^{-\varepsilon n G_{\mathcal{S}}^\rho(x)/2L}. \tag{5}$$

Note that while exactly sampling from the un-normalized density $\pi_{\mathsf{A}_{\mathrm{gr-inv}}(\mathcal{S})}$ is computationally intractable, analyzing its performance is an important step towards understanding the optimal rates for the family of functions with growth that we study in this work. The following theorem demonstrates the adaptivity of the inverse sensitivity mechanism to the growth of the underlying instance. We defer the proof to Appendix A.

**Theorem 1.** *Let $\mathcal{S} = (s_1, \ldots, s_n) \in \mathbb{S}^n$, $F(x; s)$ be convex, $L$-Lipschitz for all $s \in \mathbb{S}$. Let $x^\star = \operatorname{argmin}_{x \in \mathcal{X}} f_{\mathcal{S}}(x)$ and assume $x^\star$ is in the interior of $\mathcal{X}$. Assume that $f_S(x)$ has $\kappa$-growth (Assumption 1) with $\kappa \geq \underline{\kappa} > 1$. For $\rho > 0$, the $\rho$-smooth inverse sensitivity mechanism $\mathsf{A}_{\mathrm{gr-inv}}$ (5) is $\varepsilon$-DP, and with probability at least $1 - \beta$ the output $\hat{x} = \mathsf{A}_{\mathrm{gr-inv}}(\mathcal{S})$ has*

$$f_{\mathcal{S}}(\hat{x}) - \min_{x \in \mathcal{X}} f_{\mathcal{S}}(x) \leq \frac{1}{\lambda^{\frac{1}{\kappa-1}}} \left( \frac{2L(\log(1/\beta) + d\log(D/\rho))}{n\varepsilon} \right)^{\frac{\kappa}{\kappa-1}} + L\rho.$$

*Moreover, setting $\rho = (L/\lambda)^{\frac{1}{\kappa-1}}(d/n\varepsilon)^{\frac{\kappa}{\kappa-1}}$, we have*

$$f_{\mathcal{S}}(\hat{x}) - \min_{x \in \mathcal{X}} f_{\mathcal{S}}(x) \leq \frac{1}{\lambda^{\frac{1}{\kappa-1}}} \widetilde{O} \left( \frac{Ld}{n\varepsilon} \right)^{\frac{\kappa}{\kappa-1}}.$$

The rates of the inverse sensitivity in Theorem 1 provide two main insights regarding the landscape of the problem with growth conditions. First, these conditions allow to improve the worst-case rate $d/n\varepsilon$ to $(d/n\varepsilon)^{\frac{\kappa}{\kappa-1}}$ for pure $\varepsilon$-DP and therefore suggest a better rate $(\sqrt{d\log(1/\delta)}/n\varepsilon)^{\frac{\kappa}{\kappa-1}}$ is possible for approximate $(\varepsilon, \delta)$-DP. Moreover, the general instance-optimality guarantees of this mechanism [2] hint that these are the optimal rates for our class of functions. In the sections to come, we validate the correctness of these predictions by developing efficient algorithms that achieve these rates (for pure and approximate privacy), and prove matching lower bounds which demonstrate the optimality of these algorithms.

# 4 Efficient algorithms with optimal rates

While the previous section demonstrates that there exists algorithms that improve the rates for functions with growth, we pointed out that $\mathsf{A}_{\mathrm{gr-inv}}$ was computationally intractable in the general case. In this section, we develop efficient algorithms—e.g. that are implementable with gradient-based methods—that achieve the same convergence rates. Our algorithms build on the recent localization techniques that Feldman et al. [25] used to obtain optimal rates for DP-SCO with general convex functions. In Section 4.1, we use these techniques to develop private algorithms that achieve the optimal rates for (pure) DP-SCO with high probability, in contrast to existing results which bound the expected excess loss. These results are of independent interest.

In Section 4.2, we translate these results into convergence guarantees on privately optimizing convex functions with growth by solving a sequence of increasingly constrained SCO problems—the high-probability guarantees of Section 4.1 being crucial to our convergence analysis of these algorithms.

## 4.1 High-probability guarantees for convex DP-SCO

We first describe our algorithm (Algorithm 1) then analyze its performance under pure-DP (Proposition 1) and approximate-DP constraints (Proposition 2). Our analysis builds on novel tight generalization bounds for uniformly-stable algorithms with high probability [24]. We defer the proofs to Appendix B.

---

**Algorithm 1** Localization-based Algorithm

---

**Require:** Dataset $\mathcal{S} = (s_1, \ldots, s_n) \in \mathbb{S}^n$, constraint set $\mathcal{X}$, step size $\eta$, initial point $x_0$, privacy parameters $(\varepsilon, \delta)$;
1: Set $k = \lceil \log n \rceil$ and $n_0 = n/k$
2: **for** $i = 1$ to $k$ **do**
3:      Set $\eta_i = 2^{-4i} \eta$
4:      Solve the following ERM over $\mathcal{X}_i = \{x \in \mathcal{X} : \|x - x_{i-1}\|_2 \leq 2L\eta_i n_0\}$:

$$F_i(x) = \frac{1}{n_0} \sum_{j=1+(i-1)n_0}^{in_0} F(x; s_j) + \frac{1}{\eta_i n_0} \|x - x_{i-1}\|_2^2$$

5:      Let $\hat{x}_i$ be the output of the optimization algorithm
6:      **if** $\delta = 0$ **then**
7:          Set $\zeta_i \sim \mathsf{Lap}_d(\sigma_i)$ where $\sigma_i = 4L\eta_i \sqrt{d}/\varepsilon_i$
8:      **else if** $\delta > 0$ **then**
9:          Set $\zeta_i \sim \mathsf{N}(0, \sigma_i^2)$ where $\sigma_i = 4L\eta_i \sqrt{\log(1/\delta)}/\varepsilon$
10:      Set $x_i = \hat{x}_i + \zeta_i$
11: **return** the final iterate $x_k$

---

**Proposition 1.** *Let $\beta \leq 1/(n+d)$, $\mathsf{diam}_2(\mathcal{X}) \leq D$ and $F(x; s)$ be convex, $L$-Lipschitz for all $s \in \mathbb{S}$. Setting*

$$\eta = \frac{D}{L} \min \left( \frac{1}{\sqrt{n \log(1/\beta)}}, \frac{\varepsilon}{d \log(1/\beta)} \right)$$

*then for $\delta = 0$, Algorithm 1 is $\varepsilon$-DP and has with probability $1 - \beta$*

$$f(x) - f(x^\star) \leq LD \cdot O \left( \frac{\sqrt{\log(1/\beta)} \log^{3/2} n}{\sqrt{n}} + \frac{d \log(1/\beta) \log n}{n\varepsilon} \right).$$

Similarly, by using a different choice for the parameters and noise distribution, we have the following guarantees for approximate $(\varepsilon, \delta)$-DP.

**Proposition 2.** *Let $\beta \leq 1/(n+d)$, $\mathsf{diam}_2(\mathcal{X}) \leq D$ and $F(x; s)$ be convex, $L$-Lipschitz for all $s \in \mathbb{S}$. Setting*

$$\eta = \frac{D}{L} \min \left( \frac{1}{\sqrt{n \log(1/\beta)}}, \frac{\varepsilon}{\sqrt{d \log(1/\delta)} \log(1/\beta)} \right),$$

*then for $\delta > 0$, Algorithm 1 is $(\varepsilon, \delta)$-DP and has with probability $1 - \beta$*

$$f(x) - f(x^\star) \leq LD \cdot O \left( \frac{\sqrt{\log(1/\beta)} \log^{3/2} n}{\sqrt{n}} + \frac{\sqrt{d \log(1/\delta)} \log(1/\beta) \log n}{n\varepsilon} \right).$$

Before presenting our algorithms for functions with growth, we remark that the exact calculation of the ERM solution in step 5 of Algorithm 1 is not necessary; we chose it to clarify the main algorithmic ideas. However, as long as the returned solution $\hat{x}_i$ is sufficiently accurate, say $F_i(\hat{x}_i) - \min_{x_i^\star \in \mathcal{X}} F_i(x_i^\star) \leq \Delta$, we have that $\|\hat{x}_i - x_i^\star\|_2 \leq \sqrt{2\Delta/\lambda_i}$ where $\lambda_i = 1/(\eta_i n_0)$. This implies that as long as $\Delta \leq \frac{L}{n}$, the sensitivity of $\hat{x}_i$ is at most twice the sensitivity of the exact ERM solution $x_i^\star$ and hence multiplying the noise $\sigma_i$ by a factor of 2 is sufficient to guarantee privacy. As the set $\mathcal{X}_i$ is convex, finding sufficiently accurate $\hat{x}_i$ can be done efficiently using standard optimization methods for minimizing convex functions over convex domains.

## 4.2 Algorithms for DP-SCO with growth

Building on the algorithms of the previous section, we design algorithms that recover the rates of the inverse sensitivity mechanism for functions with growth, importantly without knowledge of the value of $\kappa$. Inspired by epoch-based algorithms from the optimization literature [31, 29], our algorithm iteratively applies the private procedures from the previous section. Crucially, the growth assumption allows to reduce the diameter of the domain after each run, hence improving the overall excess loss by carefully choosing the hyper-parameters. We provide full details in Algorithm 2.

---

**Algorithm 2** Epoch-based algorithms for $\kappa$-growth

---

**Require:** Dataset $\mathcal{S} = (s_1, \ldots, s_n) \in \mathbb{S}^n$, convex set $\mathcal{X}$, initial point $x_0$, number of iterations $T$, privacy parameters $(\varepsilon, \delta)$;

1: Set $n_0 = n/T$ and $D_0 = \mathsf{diam}_2(\mathcal{X})$
2: **if** $\delta = 0$ **then**
3:     Set $\eta_0 = \frac{D_0}{2L} \min \left( \frac{1}{\sqrt{n_0 \log(n_0) \log(1/\beta)}}, \frac{\varepsilon}{d \log(1/\beta)} \right)$
4: **else if** $\delta > 0$ **then**
5:     $\eta_0 = \frac{D_0}{2L} \min \left( \frac{1}{\sqrt{n_0 \log(n_0) \log(1/\beta)}}, \frac{\varepsilon}{\sqrt{d \log(1/\delta)} \log(1/\beta)} \right)$
6: **for** $i = 0$ to $T - 1$ **do**
7:     Let $\mathcal{S}_i = (s_{1+(i-1)n_0}, \ldots, s_{in_0})$
8:     Set $D_i = 2^{-i} D_0$ and $\eta_i = 2^{-i} \eta_0$
9:     Set $\mathcal{X}_i = \{x \in \mathcal{X} : \|x - x_i\|_2 \leq D_i\}$
10:     Run Algorithm 1 on dataset $\mathcal{S}_i$ with starting point $x_i$, privacy parameter $(\varepsilon, \delta)$, domain $\mathcal{X}_i$ (with diameter $D_i$), step size $\eta_i$
11:     Let $x_{i+1}$ be the output of the private procedure
12: **return** $x_T$

---

The following theorem summarizes our main upper bound for DP-SCO with growth in the pure privacy model, recovering the rates of the inverse sensitivity mechanism in Section 3. We defer the proof to Appendix B.3.

**Theorem 2.** *Let $\beta \leq 1/(n+d)$, $\mathsf{diam}_2(\mathcal{X}) \leq D$ and $F(x; s)$ be convex, L-Lipschitz for all $s \in \mathbb{S}$. Assume that $f$ has $\kappa$-growth (Assumption 1) with $\kappa \geq \underline{\kappa} > 1$. Setting $T = \left\lceil \frac{2 \log n}{\underline{\kappa} - 1} \right\rceil$, Algorithm 2 is $\varepsilon$-DP and has with probability $1 - \beta$*

$$f(x_T) - \min_{x \in \mathcal{X}} f(x) \leq \frac{1}{\lambda^{\frac{1}{\kappa - 1}}} \cdot \widetilde{O} \left( \frac{L\sqrt{\log(1/\beta)}}{\sqrt{n}} + \frac{Ld \log(1/\beta)}{n\varepsilon(\underline{\kappa} - 1)} \right)^{\frac{\kappa}{\kappa - 1}},$$

*where $\widetilde{O}$ hides logarithmic factors depending on $n$ and $d$.*

*Sketch of the proof.* The main challenge of the proof is showing that the iterate achieves good risk without knowledge of $\kappa$. Let us denote by $D \cdot \rho$ the error guarantee of Proposition 1 (or Proposition 2 for approximate-DP). At each stage $i$, as long as $x^\star = \arg\min_{x \in \mathcal{X}} f(x)$ belongs to $\mathcal{X}_i$, the excess loss is of order $D_i \cdot \rho$ and thus decreases exponentially fast with $i$. The challenge is that, without knowledge of $\kappa$, we do not know the index $i_0$ (roughly $\frac{\log_2 n}{\kappa - 1}$) after which $x^\star \notin D_j$ for $j \geq i_0$ and the regret guarantees become meaningless with respect to the original problem. However, in the stages after $i_0$, as the constraint set becomes very small, we upper bound the variations in function values $f(x_{j+1}) - f(x_j)$ and show that the sub-optimality cannot increase (overall) by more than $O(D_{i_0} \cdot \rho)$, thus achieving the optimal rate of stage $i_0$.

$\square$

Moreover, we can improve the dependence on the dimension for approximate $(\varepsilon, \delta)$-DP, resulting in the following bounds.

**Theorem 3.** *Let $\beta \leq 1/(n+d)$, $\mathsf{diam}_2(\mathcal{X}) \leq D$ and $F(x; s)$ be convex, L-Lipschitz for all $s \in \mathbb{S}$. Assume that $f$ has $\kappa$-growth (Assumption 1) with $\kappa \geq \underline{\kappa} > 1$. Setting $T = \left\lceil \frac{2 \log n}{\underline{\kappa} - 1} \right\rceil$ and $\delta > 0$,*

*Algorithm 2 is $(\varepsilon, \delta)$-DP and has with probability $1 - \beta$*

$$f(x_T) - \min_{x \in \mathcal{X}} f(x) \leq \frac{1}{\lambda^{\frac{1}{\kappa-1}}} \cdot \widetilde{O}\left( \frac{L\sqrt{\log(1/\beta)}}{\sqrt{n}} + \frac{L\sqrt{d\log(1/\delta)}\log(1/\beta)}{n\varepsilon(\underline{\kappa} - 1)} \right)^{\frac{\kappa}{\kappa-1}},$$

*where $\widetilde{O}$ hides logarithmic factors depending on $n$ and $d$.*

## 5 Lower bounds

In this section, we develop (minimax) lower bounds for the problem of SCO with $\kappa$-growth under privacy constraints. Note that taking $\varepsilon \to \infty$ provides lower bound for the unconstrained minimax risk. For a sample space $\mathbb{S}$ and collection of distributions $\mathcal{P}$ over $\mathbb{S}$, we define the function class $\mathcal{F}^\kappa(\mathcal{P})$ as the set of convex functions from $\mathbb{R}^d \to \mathbb{R}$ that are $L$-Lipschitz and has $\kappa$-growth (Assumption 1). We define the *constrained* minimax risk [6]

$$\mathfrak{M}_n(\mathcal{X}, \mathcal{P}, \mathcal{F}^\kappa, \varepsilon, \delta) := \inf_{\widehat{x}_n \in \mathcal{A}^{\varepsilon, \delta}} \sup_{(F, P) \in \mathcal{F}^\kappa \times \mathcal{P}} \mathbb{E}\left[ f(\widehat{x}_n(S_1^n)) - \inf_{x' \in \mathcal{X}} f(x') \right], \qquad (6)$$

where $\mathcal{A}^{\epsilon, \delta}$ is the collection of $(\varepsilon, \delta)$-DP mechanisms from $\mathbb{S}^n$ to $\mathcal{X}$. When clear from context, we omit the dependency on $\mathcal{P}$ of the function class and simply write $\mathcal{F}^\kappa$. We also forego the dependence on $\delta$ when referring to pure-DP constraints, i.e. $\mathfrak{M}_n(\mathcal{X}, \mathcal{P}, \mathcal{F}^\kappa, \varepsilon, \delta = 0) =: \mathfrak{M}_n(\mathcal{X}, \mathcal{P}, \mathcal{F}^\kappa, \varepsilon)$. We now proceed to prove tight lower bounds for $\varepsilon$-DP in Section 5.1 and $(\varepsilon, \delta)$-DP in Section 5.2.

### 5.1 Lower bounds for pure $\varepsilon$-DP

Although in Section 4 we show that the same algorithm achieves the optimal upper bounds for all values of $\kappa > 1$, the landscape of the problem is more subtle for the lower bounds and we need to delineate two different cases to obtain tight lower bounds. We begin with $\kappa \geq 2$, which corresponds to uniform convexity and enjoys properties that make the problem easier (e.g., closure under summation or addition of linear terms). The second case, $1 < \kappa < 2$, corresponds to sharper growth and requires a different hard instance to satisfy the growth condition.

**$\kappa$-growth with $\kappa \geq 2$.** We begin by developing lower bounds under pure DP for $\kappa \geq 2$

**Theorem 4** (Lower bound for $\varepsilon$-DP, $\kappa \geq 2$). *Let $d \geq 1$, $\mathcal{X} = \mathbb{B}_2^d(R)$, $\mathbb{S} = \{\pm e_j\}_{j \leq d}$, $\kappa \geq 2$ and $n \in \mathbb{N}$. Let $\mathcal{P}$ be the set of distributions on $\mathbb{S}$. Assume that*

$$2^{\kappa-1} \leq \frac{L}{\lambda} \frac{1}{R^{\kappa-1}} \leq 2^{\kappa-1}\sqrt{96n} \ \text{ and } \ n\varepsilon \geq \frac{1}{\sqrt{3}}$$

*The following lower bound holds*

$$\mathfrak{M}_n(\mathcal{X}, \mathcal{P}, \mathcal{F}^\kappa, \epsilon) \geq \frac{1}{\lambda^{\frac{1}{\kappa-1}}} \widetilde{\Omega}\left( \left( \frac{L}{\sqrt{n}} \right)^{\frac{\kappa}{(\kappa-1)}} + \left( \frac{Ld}{n\varepsilon} \right)^{\frac{\kappa}{\kappa-1}} \right). \qquad (7)$$

First of all, note that $L \geq \lambda 2^\kappa R^{\kappa-1}$ is not an overly-restrictive assumption. Indeed, for an arbitrary $(\lambda, \kappa)$-uniformly convex and $L$-Lipschitz function, it always holds that $L \geq \frac{\lambda}{2} R^{\kappa-1}$. This is thus equivalent to assuming $\kappa = \Theta(1)$. Note that when $\kappa \gg 1$, the standard $n^{-1/2} + d/(n\varepsilon)$ lower bound holds. We present the proof in Appendix C.1.1 and preview the main ideas here.

*Sketch of the proof.* Our lower bounds hinges on the collections of functions $F(x; s) := a\kappa^{-1}\|x\|_2^\kappa + b\langle x, s \rangle$ for $a, b \geq 0$ to be chosen later. These functions are [39, Lemma 4] $\kappa$-uniformly convex for any $s \in \mathbb{S}$ and in turn, so is the population function $f$. We proceed as follows, we first prove an information-theoretic (non-private) lower bound (Theorem 8 in Appendix C.1.1) which provides the statistical term in (7). With the same family of functions, we exhibit a collection of datasets and prove by contradiction that if an estimator were to optimize below a certain error it would have violated $\varepsilon$-DP—this yields a lower bound on ERM for our function class (Theorem 9 in Appendix C.1.1). We conclude by proving a reduction from SCO to ERM in Proposition 4. $\qquad \square$

$\kappa$**-growth with** $\kappa \in (1, 2]$**.** As the construction of the hard instance is more intricate for $\kappa < 2$, we provide a one-dimensional lower bound and leave the high-dimensional case for future work. In this case we directly obtain the result with a private version of Le Cam's method [44, 42, 6], however with a different family of functions.

The issue with the construction of the previous section is that the function does not exhibit sharp growth for $\kappa < 2$. Indeed, the added linear function shifts the minimum away from 0 where the function is differentiable and as a result it locally behaves as a quadratic and only achieves growth $\kappa = 2$. To establish the lower bound, we consider a different sample function $F$ that has growth exactly 1 on one side and $\kappa$ on the other side. This yields the following

**Theorem 5** (Lower bound for $\varepsilon$-DP, $\kappa \in (1, 2]$ )**.** *Let* $d = 1$, $\mathbb{S} = \{-1, +1\}$, $\kappa \in (1, 2]$, $\lambda = 1$, $L = 2$, *and* $n \in \mathbb{N}$. *There exists a collection of distributions* $\mathcal{P}$ *such that, whenever* $n\varepsilon \geq 1/\sqrt{3}$, *it holds that*

$$\mathfrak{M}_n([-1, 1], \mathcal{P}, \mathcal{F}^\kappa_{d=1}, \epsilon) = \Omega\left\{ \left(\frac{1}{\sqrt{n}}\right)^{\frac{\kappa}{\kappa-1}} + \left(\frac{1}{n\varepsilon}\right)^{\frac{\kappa}{\kappa-1}} \right\}. \tag{8}$$

## 5.2 Lower bounds under approximate privacy constraints

We conclude our treatment by providing lower bounds but now under *approximate* privacy constraints, demonstrating the optimality of the risk bound of Theorem 3. We prove the result via a reduction: we show that if one solves ERM with $\kappa$-growth with error $\Delta$, this implies that one solves arbitrary convex ERM with error $\phi(\Delta)$. Given that a lower bound of $\Omega(\sqrt{d}/(n\varepsilon))$ holds for ERM, a lower bound of $\phi^{-1}(\sqrt{d}/(n\varepsilon))$ holds for ERM with $\kappa$-growth. However, for this reduction to hold, we require that $\kappa \geq 2$. Furthermore, we consider $\kappa$ to be roughly a constant—in the case that $\kappa$ is too large, standard lower bounds on general convex functions apply.

**Theorem 6** (Private lower bound for $(\varepsilon, \delta)$-DP)**.** *Let* $\kappa \geq 2$ *such that* $\kappa = \Theta(1)$, $\mathcal{X} = \mathbb{B}^d_2(D)$. *Let* $d \geq 1$ *and* $\mathbb{S} = \{\pm 1/\sqrt{d}\}^d$. *Assume that* $n\varepsilon = \Omega(\sqrt{d})$, *then for any* $(\varepsilon, \delta)$ *mechanism* $\mathsf{A}$, *there exists* $\lambda > 0$, $F$ *and* $\mathcal{S} \subset \mathbb{S}$ *such that*

$$\mathbb{E}[f_\mathcal{S}(\mathsf{A}(\mathcal{S}))] - \inf_{x' \in \mathcal{X}} f_\mathcal{S}(x') \geq \tilde{\Omega}\left[ \frac{1}{\lambda^{\frac{1}{\kappa-1}}} \left(\frac{L\sqrt{d}}{n\varepsilon}\right)^{\frac{\kappa}{\kappa-1}} \right].$$

Theorem 6 implies that the same lower bound (up to logarithmic factors) applies to SCO via the reduction of [8, Appendix C]. Before proving the theorem, let us state (and prove in Appendix C.2) the following reduction: if an $(\varepsilon, \delta)$-DP algorithm achieves excess error (roughly) $\Delta$ on ERM for any function with $\kappa$-growth, there exists an $(\varepsilon, \delta)$-DP algorithm that achieves error $\Delta^{(\kappa-1)/\kappa}$ for any convex function. We construct the latter by iteratively solving ERM problems with geometrically increasing $\| \cdot \|^\kappa_2$-regularization towards the previous iterate to ensure the objective has $\kappa$-growth.

**Proposition 3** (Solving ERM with $\kappa$-growth implies solving any convex ERM)**.** *Let* $\kappa \geq 2$. *Assume there exists an* $(\epsilon, \delta)$ *mechanism* $\mathsf{A}$ *such that for any* $L$*-Lipschitz loss* $G$ *on* $\mathcal{Y}$ *and dataset* $\mathcal{S}$ *such that* $g_\mathcal{S}(x) := \frac{1}{n}\sum_{s \in \mathcal{S}} G(x; s)$ *exhibits* $(\lambda, \kappa)$*-growth, the mechanism achieves excess loss*

$$\mathbb{E}[g_\mathcal{S}(\mathsf{A}(\mathcal{S}, G, \mathcal{Y}))] - \inf_{y' \in \mathcal{Y}} g_\mathcal{S}(y') \leq \frac{1}{\lambda^{\frac{1}{\kappa-1}}}\Delta(n, L, \epsilon, \delta).$$

*Then, we can construct an* $(\varepsilon, \delta)$*-DP mechanism* $\mathsf{A}'$ *such that for any* $L$*-Lipschitz loss* $f$, *the mechanism achieves excess loss*

$$\mathbb{E}[f_\mathcal{S}(\mathsf{A}'(\mathcal{S}))] - \inf_{x' \in \mathcal{X}} f_\mathcal{S}(x') \leq O\left( D[\Delta(n, L, \epsilon/k, \delta/k)]^{\frac{\kappa-1}{\kappa}} \right),$$

*where $k$ is the smallest integer such that* $k \geq \log\left[ \frac{\kappa^{\frac{1}{\kappa-1}} L^{\frac{\kappa}{\kappa-1}}}{2^{2\kappa-3}\Delta(n, L, \varepsilon/k, \delta/k)} \right]$.

With this proposition, the proof of the theorem directly follows as Bassily et al. [7] prove a lower bound $\Omega(\sqrt{d}/(n\varepsilon))$ for ERM with $(\varepsilon, \delta)$-DP.

## Discussion

In this work, we develop private algorithms that adapt to the growth of the function at hand, achieving the convergence rate corresponding to the "easiest" sub-class the function belongs to. However, the picture is not yet complete. First of, there are still gaps in our theoretical understanding, the most interesting one being $\kappa = 1$. On these functions, appropriate optimization algorithms achieve linear convergence [43] and raise the question, can we achieve exponentially small privacy cost in our setting? Finally, while our optimality guarantees are more fine-grained than the usual minimax results over convex functions, they are still contigent on some predetermined choice of sub-classes. Studying more general notions of adaptivity is an important future direction in private optimization.

### Acknowledgments

The authors would like to thank Karan Chadha and Gary Cheng for comments on an early version of the draft. HA, DL and JCD were supported NSF under CAREER Award CCF-1553086 and HDR 1934578 (Stanford Data Science Collaboratory), Office of Naval Research YIP Award N00014-19-2288 and the Stanford DAWN Consortium.

### Competing Interests

JCD has a consulting relationship with Apple. HS has spent internships at Apple during the 2019, 2020 and 2021 summers. DL has spent an internship at Google during the summer of 2020.

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
