## Potential negative societal impact

The aim of our work is theoretical in essence and as such, we do not expect direct negative societal impact. As DP becomes a more established norm, we believe this research is relevant for practitionners in both industry and government. Indeed, an important obstacle to applying DP is the loss of performance compared to non-private models; our theoretical results suggests that better adaptive algorithms would significantly narrow this performance gap. We wish to point out two potential negative consequences of growing research in privacy. First, a simple but effective method to guarantee privacy is to either delete existing user data or limit data collection in the first place. Paradoxically, the more confident institutions are in DP algorithms, the less they are susceptible to turn to these simpler—and most effective—solutions. Finally, using DP algorithms should not preclude one from (1) carefully choosing $\epsilon$ and $\delta$ to provide meaningful guarantees for the specific application at hand and (2) developing exhaustive and meticulous evaluation methods of the privacy of deployed models.

## A  Proofs for Section 3

### A.1  Proof of Theorem 1

**Theorem 1.** *Let $\mathcal{S} = (s_1, \ldots, s_n) \in \mathbb{S}^n$, $F(x; s)$ be convex, $L$-Lipschitz for all $s \in \mathbb{S}$. Let $x^\star = \mathrm{argmin}_{x \in \mathcal{X}} f_{\mathcal{S}}(x)$ and assume $x^\star$ is in the interior of $\mathcal{X}$. Assume that $f_S(x)$ has $\kappa$-growth (Assumption 1) with $\kappa \geq \underline{\kappa} > 1$. For $\rho > 0$, the $\rho$-smooth inverse sensitivity mechanism $\mathsf{A}_{\mathrm{gr-inv}}$ (5) is $\varepsilon$-DP, and with probability at least $1 - \beta$ the output $\hat{x} = \mathsf{A}_{\mathrm{gr-inv}}(\mathcal{S})$ has*

$$f_{\mathcal{S}}(\hat{x}) - \min_{x \in \mathcal{X}} f_{\mathcal{S}}(x) \leq \frac{1}{\lambda^{\frac{1}{\kappa-1}}} \left( \frac{2L(\log(1/\beta) + d\log(D/\rho))}{n\varepsilon} \right)^{\frac{\kappa}{\kappa-1}} + L\rho.$$

*Moreover, setting $\rho = (L/\lambda)^{\frac{1}{\kappa-1}}(d/n\varepsilon)^{\frac{\kappa}{\kappa-1}}$, we have*

$$f_{\mathcal{S}}(\hat{x}) - \min_{x \in \mathcal{X}} f_{\mathcal{S}}(x) \leq \frac{1}{\lambda^{\frac{1}{\kappa-1}}} \widetilde{O} \left( \frac{Ld}{n\varepsilon} \right)^{\frac{\kappa}{\kappa-1}}.$$

Let us first prove privacy. The sensitivity of $\|\nabla f_{\mathcal{S}}(x)\|_2$ is $2L/n$ as $F$ is $L$-Lipschitz, therefore following the privacy proof of the smooth inverse sensitivity mechanism [2, Prop. 3.2] we get that $\mathsf{A}_{\mathrm{gr-inv}}$ (5) is $\varepsilon$-DP.

Let us now prove the claim about utility. Denote $\hat{x} = \mathsf{A}_{\mathrm{gr-inv}}(\mathcal{S})$ and $E = \frac{2LK}{n\varepsilon}$ with $K$ to be chosen presently. We argue that it is enough to show that $\Pr(G_\rho(\hat{x}) \geq E) \leq \beta$. Indeed then with probability at least $1 - \beta$ we have $G_\rho(\hat{x}) \leq E$, which implies there is $y$ such that $\|\hat{x} - y\|_2 \leq \rho$ and $\|\nabla f_{\mathcal{S}}(y)\|_2 \leq E$, hence using the Kurdyka-Łojasiewicz inequality (2)

$$
\begin{aligned}
f_{\mathcal{S}}(\hat{x}) - f_{\mathcal{S}}(x^\star) &= f_{\mathcal{S}}(\hat{x}) - f_{\mathcal{S}}(y) + f_{\mathcal{S}}(y) - f_{\mathcal{S}}(x^\star) \\
&\leq L\rho + \frac{e}{\lambda^{\frac{1}{\kappa-1}}} \|\nabla f_{\mathcal{S}}(y)\|_2^{\frac{\kappa}{\kappa-1}} \\
&\leq L\rho + \frac{e}{\lambda^{\frac{1}{\kappa-1}}} E^{\frac{\kappa}{\kappa-1}}.
\end{aligned}
$$

It remains to prove that $\Pr(G_\rho(\hat{x}) \geq E) \leq \beta$. Let $S_0 = \{x \in \mathbb{R}^d : \|x - x^\star\|_2 \leq \rho\}$ and $S_1 = \{x \in \mathbb{R}^d : G_\rho(x) \geq E\}$. Note that $G_\rho(x) = 0$ for any $x \in S_0$ as $x^\star$ is in the interior of $\mathcal{X}$ which implies $\nabla f_{\mathcal{S}}(x^\star) = 0$. Hence the definition of the smooth inverse sensitivity mechanism (5) implies

$$
\begin{aligned}
\Pr(\mathsf{A}_{\mathrm{gr-inv}}(\mathcal{S}) \in S_1) &\leq \frac{\mathsf{Vol}(\{x \in \mathbb{R}^d : \|x - x^\star\|_2 \leq D + \rho\})e^{-\frac{n\varepsilon}{2L}E}}{\mathsf{Vol}(\{x \in \mathbb{R}^d : \|x - x^\star\|_2 \leq \rho\})} \\
&\leq e^{-K}\left(1 + \frac{D}{\rho}\right)^d \leq \beta,
\end{aligned}
$$

where the last inequality follows by choosing $K = \log(1/\beta) + d\log(1 + D/\rho)$.

# B  Proofs for Section 4

We need to the following result on the generalization properties of uniformly stable algorithms [24].

**Theorem 7.** *[24, Cor. 4.2] Assume* $\mathsf{diam}_2(\mathcal{X}) \leq D$. *Let* $\mathcal{S} = (S_1, \ldots, S_n)$ *where* $S_1^n \overset{\text{iid}}{\sim} P$ *and* $F(x; s)$ *is* $L$-*Lipschitz and* $\lambda$-*strongly convex for all* $s \in \mathbb{S}$. *Let* $\hat{x} = \operatorname{argmin}_{x \in \mathcal{X}} f_{\mathcal{S}}(x)$ *be the empirical minimizer. For* $0 < \beta \leq 1/n$, *with probability at least* $1 - \beta$

$$f(\hat{x}) - f(x^\star) \leq \frac{cL^2 \log(n) \log(1/\beta)}{\lambda n} + \frac{cLD\sqrt{\log(1/\beta)}}{\sqrt{n}}.$$

## B.1  Proof of Proposition 1

**Proposition 1.** *Let* $\beta \leq 1/(n + d)$, $\mathsf{diam}_2(\mathcal{X}) \leq D$ *and* $F(x; s)$ *be convex,* $L$-*Lipschitz for all* $s \in \mathbb{S}$. *Setting*

$$\eta = \frac{D}{L} \min \left( \frac{1}{\sqrt{n \log(1/\beta)}}, \frac{\varepsilon}{d \log(1/\beta)} \right)$$

*then for* $\delta = 0$, *Algorithm 1 is* $\varepsilon$-*DP and has with probability* $1 - \beta$

$$f(x) - f(x^\star) \leq LD \cdot O \left( \frac{\sqrt{\log(1/\beta)} \log^{3/2} n}{\sqrt{n}} + \frac{d \log(1/\beta) \log n}{n\varepsilon} \right).$$

We begin by proving the privacy claim. We show that each iterate is $\varepsilon$-DP and therefore post-processing implies the claim as each sample is used in exactly one iterate. To this end, let $\lambda_i = 1/\eta_i n_0$ and note that the minimizer $\hat{x}_i$ has $\ell_2$ sensitivity $2L/\lambda_i n_0 \leq 4L\eta_i$ [25], hence the $\ell_1$-sensitivity is at most $4L\eta_i \sqrt{d}$. Standard properties of the Laplace mechanism [20] now imply that $x_i$ is $\varepsilon$-DP which give the claim about privacy.

Now we proceed to prove utility which follows similar arguments to the localization-based proof in [25]. Letting $\hat{x}_0 = x^\star$, we have:

$$f(x_k) - f(x^\star) = \sum_{i=1}^{k} f(\hat{x}_i) - f(\hat{x}_{i-1}) + f(x_k) - f(\hat{x}_k).$$

First, by using standard properties of Laplace distributions [17], we know that for $\zeta_i \sim \mathsf{Lap}(\sigma_i)$,

$$\Pr(\|\zeta_i\|_2 \geq t) \leq \Pr(\|\zeta_i\|_\infty \geq t/\sqrt{d}) \leq d e^{-t/\sqrt{d}\sigma_i},$$

which implies (as $\beta \leq 1/(n+d)$) that with probability $1 - \beta/2$ we have $\|\zeta_i\|_2 \leq 10\sqrt{d}\sigma_i \log(1/\beta)$ for all $1 \leq i \leq k$. Hence

$$
\begin{aligned}
f(x_k) - f(\hat{x}_k) &\leq L\|x_k - \hat{x}_k\|_2 \\
&\leq L\sigma_k \sqrt{d} \log(1/\beta) \\
&\leq 4L^2 d \frac{\eta_i}{\varepsilon} \\
&\leq 4L^2 d \frac{\eta}{\varepsilon 2^{4i}} \leq \frac{4LD}{n^2},
\end{aligned}
$$

where the last inequality follows since $\eta = \frac{D\varepsilon}{Ld \log(k/\beta)}$. Now we use high-probability generalization guarantees of uniformly-stable algorithms. We use Theorem 7 with $F(x; s_j) + \frac{\|x - x_{i-1}\|_2^2}{\eta_i n_0}$ to get that with probability $1 - \beta/2$ for each $i$

$$f(\hat{x}_i) - f(\hat{x}_{i-1}) \leq \frac{\|\hat{x}_{i-1} - x_{i-1}\|_2^2}{\eta_i n_0} + cL^2 \log(n) \log(1/\beta)\eta_i + \frac{cLD\sqrt{\log(1/\beta)}}{\sqrt{n_0}}.$$

Thus,

$$\sum_{i=1}^{k} f(\hat{x}_i) - f(\hat{x}_{i-1}) \le \sum_{i=1}^{k} \left\{ \frac{\|\hat{x}_{i-1} - x_{i-1}\|_2^2}{\eta_i n_0} + cL^2 \log(n) \log(1/\beta) \eta_i + \frac{cLD\sqrt{\log(1/\beta)}}{\sqrt{n_0}} \right\}$$

$$\le \frac{D^2}{\eta n_0} + \left[ \sum_{i=2}^{k} \frac{\sigma_{i-1}^2 d \log^2(1/\beta)}{\eta_i n_0} \right] + 2cL^2 \log(n) \log(1/\beta) \eta + \frac{cLD\sqrt{\log(1/\beta)}k}{\sqrt{n_0}}$$

$$= \frac{D^2}{\eta n_0} + \left[ \sum_{i=2}^{k} \frac{CL^2 \eta_{i-1} d^2 \log^2(1/\beta)}{n_0 \varepsilon^2} \right] + 2cL^2 \log(n) \log(1/\beta) \eta + \frac{cLD\sqrt{\log(1/\beta)}k}{\sqrt{n_0}}$$

$$= \frac{D^2}{\eta n_0} + \frac{CL^2 \eta d^2 \log^2(1/\beta)}{n_0 \varepsilon^2} \left[ \sum_{i=2}^{k} 2^{-i} \right] + 2cL^2 \log(n) \log(1/\beta) \eta + \frac{cLD\sqrt{\log(1/\beta)}k}{\sqrt{n_0}}$$

$$\le LD \cdot O \left( \frac{\sqrt{\log(1/\beta)\log(n)} + \sqrt{\log(1/\beta)} \log^{3/2}(n)}{\sqrt{n}} + \frac{d \log(1/\beta) \log(n)}{n\varepsilon} \right),$$

where the last inequality follows by choosing $\eta = \frac{D}{L} \min \left( \frac{1}{\sqrt{n \log(1/\beta)}}, \frac{\varepsilon}{d \log(1/\beta)} \right)$

## B.2 Proof of Proposition 2

**Proposition 2.** *Let $\beta \le 1/(n+d)$, $\mathsf{diam}_2(\mathcal{X}) \le D$ and $F(x; s)$ be convex, L-Lipschitz for all $s \in \mathbb{S}$. Setting*

$$\eta = \frac{D}{L} \min \left( \frac{1}{\sqrt{n \log(1/\beta)}}, \frac{\varepsilon}{\sqrt{d \log(1/\delta)} \log(1/\beta)} \right),$$

*then for $\delta > 0$, Algorithm 1 is $(\varepsilon, \delta)$-DP and has with probability $1 - \beta$*

$$f(x) - f(x^\star) \le LD \cdot O \left( \frac{\sqrt{\log(1/\beta)} \log^{3/2} n}{\sqrt{n}} + \frac{\sqrt{d \log(1/\delta)} \log(1/\beta) \log n}{n\varepsilon} \right).$$

The proof is similar to the proof of Proposition 1. For privacy, we show in the proof of Proposition 1 that the $\ell_2$-sensitivity of $\hat{x}_i$ is upper bounded by $2L/\lambda_i n_0 \le 4L\eta_i$ hence standard properties of the Gaussian mechanism [20] now imply that $x_i$ is $(\varepsilon, \delta)$-DP which implies the final algorithm is $(\varepsilon, \delta)$-DP using post-processing.

The utility proof follows the same arguments as in the proof of Proposition 1, except that for $\zeta_i \sim N(0, \sigma_i^2)$ we have [30] (since $\zeta_i$ is $2\sqrt{2}\sigma_i\sqrt{d}$-norm-sub-Gaussian)

$$\Pr(\|\zeta_i\|_2 \ge t\sqrt{d}) \le 2e^{-\frac{t^2}{16\sigma_i^2}},$$

implying that $\|\zeta_i\|_2 \le 4\sqrt{d}\sigma_i \log(4/\beta)$ for all $1 \le i \le k$ with probability $1 - \beta/2$.

## B.3 Proofs of Theorems 2 and 3

We first restate Theorems 2 and 3.

**Theorem 2.** *Let $\beta \le 1/(n+d)$, $\mathsf{diam}_2(\mathcal{X}) \le D$ and $F(x; s)$ be convex, L-Lipschitz for all $s \in \mathbb{S}$. Assume that $f$ has $\kappa$-growth (Assumption 1) with $\kappa \ge \underline{\kappa} > 1$. Setting $T = \left\lceil \frac{2 \log n}{\underline{\kappa} - 1} \right\rceil$, Algorithm 2 is $\varepsilon$-DP and has with probability $1 - \beta$*

$$f(x_T) - \min_{x \in \mathcal{X}} f(x) \le \frac{1}{\lambda^{\frac{1}{\kappa-1}}} \cdot \widetilde{O} \left( \frac{L\sqrt{\log(1/\beta)}}{\sqrt{n}} + \frac{Ld \log(1/\beta)}{n\varepsilon(\underline{\kappa} - 1)} \right)^{\frac{\kappa}{\kappa-1}},$$

*where $\widetilde{O}$ hides logarithmic factors depending on $n$ and $d$.*

**Theorem 3.** *Let $\beta \leq 1/(n + d)$, $\mathrm{diam}_2(\mathcal{X}) \leq D$ and $F(x; s)$ be convex, $L$-Lipschitz for all $s \in \mathbb{S}$. Assume that $f$ has $\kappa$-growth (Assumption 1) with $\kappa \geq \underline{\kappa} > 1$. Setting $T = \left\lceil \frac{2 \log n}{\underline{\kappa} - 1} \right\rceil$ and $\delta > 0$, Algorithm 2 is $(\varepsilon, \delta)$-DP and has with probability $1 - \beta$*

$$f(x_T) - \min_{x \in \mathcal{X}} f(x) \leq \frac{1}{\lambda^{\frac{1}{\kappa - 1}}} \cdot \widetilde{O}\left( \frac{L\sqrt{\log(1/\beta)}}{\sqrt{n}} + \frac{L\sqrt{d \log(1/\delta)} \log(1/\beta)}{n \varepsilon (\underline{\kappa} - 1)} \right)^{\frac{\kappa}{\kappa - 1}},$$

*where $\widetilde{O}$ hides logarithmic factors depending on $n$ and $d$.*

We start by proving privacy. Since each sample $s_i$ is used in exactly one iterate, we only need to show that each iterate is $(\varepsilon, \delta)$-DP, which will imply the main claim using post-processing. The privacy of each iterate follows directly from the privacy guarantees of Algorithm 1. We proceed to prove utility.

We will prove the utility claim assuming the subroutine used in Algorithm 2 satisfies the following: the output $x_{k+1}$ has error

$$f(x_{k+1}) - \min_{x \in \mathcal{X}} f(x) \leq D_k \cdot \rho,$$

for some $\rho > 0$. Note that in our setting, Proposition 1 implies that $\rho \leq L \cdot O(\frac{\sqrt{\log(1/\beta)} \log n_0}{\sqrt{n_0}} + \frac{d \log(1/\beta)}{n_0 \varepsilon})$ for pure-DP and similarly Proposition 2 gives the corresponding $\rho$ for $(\varepsilon, \delta)$-DP.

The proof has two stages. In the first stage (Lemma B.1), we prove that as long as $i \leq i_0$ for some $i_0 > 0$, then $x^\star \in \mathcal{X}_i$ and the performance of the algorithm keeps improving. We show that at the end of this stage, the points $x_{i_0+1}$ has optimal excess loss. Then, in the second stage (Lemma B.2), we show that the iterates would not move much as the radius $D_i$ of the domain is sufficiently small, hence the final accumulated error along these iterations is small.

Let us begin with the first stage. Let $i_0$ be the largest $i$ such that $D_i \geq (\frac{\kappa 2^\kappa \rho}{\lambda})^{\frac{1}{\kappa - 1}}$. We prove that $x^\star \in \mathcal{X}_i$ for all $0 \leq i \leq i_0$ where we recall that $\mathcal{X}_i = \{x \in \mathcal{X} : \|x - x_i\|_2 \leq D_i\}$ and $D_i = 2^{-i} D_0$.

**Lemma B.1.** *For all $0 \leq i \leq i_0$ we have*

$$x^\star \in \mathcal{X}_i \quad and \quad f(x_{i_0+1}) - \min_{x \in \mathcal{X}} f(x) \leq 6(2^\kappa)^{\frac{1}{\kappa - 1}} \frac{1}{\lambda^{\frac{1}{\kappa - 1}}} \rho^{\frac{\kappa}{\kappa - 1}}.$$

*Proof.* To prove the first part, we need to show that $\|x_i - x^\star\|_2 \leq D_i$. Let $\bar{D}_i = \|x_i - x^\star\|_2$. First, note that the claim is true for $i = 0$. Now we assume it is correct for $0 \leq i \leq i_0 - 1$ and prove correctness for $i + 1$. Note that the growth condition implies

$$\bar{D}_{i+1} \leq (\kappa \Delta_i / \lambda)^{1/\kappa},$$

where $\Delta_i = f(x_{i+1}) - \min_{x \in \mathcal{X}} f(x) \leq D_i \cdot \rho$. Thus we have

$$\bar{D}_{i+1} \leq (\kappa D_i \rho / \lambda)^{1/\kappa} \leq D_i / 2 = D_{i+1},$$

where the second inequality holds for $i$ that satisfies $D_i \geq (\frac{\kappa 2^\kappa \rho}{\lambda})^{\frac{1}{\kappa - 1}}$. This proves the first part of the claim. For the second part, note that the definition of $i_0$ implies that $D_{i_0} \leq 2(\frac{\kappa 2^\kappa \rho}{\lambda})^{\frac{1}{\kappa - 1}}$. Therefore, as $x^\star \in \mathcal{X}_{i_0}$ and the algorithm has error $D_i \cdot \rho$, we have

$$f(x_{i_0+1}) - \min_{x \in \mathcal{X}} f(x) \leq D_{i_0} \cdot \rho$$

$$\leq 2(\kappa 2^\kappa / \lambda)^{\frac{1}{\kappa - 1}} \rho^{\frac{\kappa}{\kappa - 1}}.$$

The claim now follows as $\kappa^{\frac{1}{\kappa - 1}} \leq 3$. $\qquad \square$

We now proceed to the second stage. The following lemma shows that the accumulated error along the iterates $i > i_0$ is small and therefore $x_T$ obtains the same error as $x_{i_0+1}$ (up to constant factors).

**Lemma B.2.** *Assume the algorithm has error $D_i \cdot \rho$. Let $i_0$ be the largest $i$ such that $D_i \geq (\frac{\kappa 2^\kappa \rho}{\lambda})^{\frac{1}{\kappa-1}}$. For all $i \geq i_0 + 1$ we have*

$$f(x_{i+1}) - f(x_i) \leq 2^{-(i-i_0)} D_{i_0} \rho.$$

*In particular, for $T \geq i_0 + 1$ we have*

$$f(x_T) - \min_{x \in \mathcal{X}} f(x) \leq 12(2^\kappa/\lambda)^{\frac{1}{\kappa-1}} \rho^{\frac{\kappa}{\kappa-1}}.$$

*Proof.* Note that as $x_i \in \mathcal{X}_i$, the guarantees of the algorithm give

$$f(x_{i+1}) - f(x_i) \leq D_i \rho = 2^{-(i-i_0)} D_{i_0} \rho.$$

For the second part of the claim, we have

$$f(x_T) - \min_{x \in \mathcal{X}} f(x) = f(x_{i_0+1}) - \min_{x \in \mathcal{X}} f(x) + \sum_{i=i_0+1}^{T} f(x_{i+1}) - f(x_i)$$

$$\leq D_{i_0} \rho + \sum_{i=i_0+1}^{T} 2^{-(i-i_0)} D_{i_0} \rho \leq 2 D_{i_0} \rho.$$

The claim now follows as $D_{i_0} \leq 2(\frac{\kappa 2^\kappa \rho}{\lambda})^{\frac{1}{\kappa-1}}$ and $\kappa^{\frac{1}{\kappa-1}} \leq 3$. $\qquad\square$

Assuming $T \geq i_0+1$, Theorem 2 and Theorem 3 now follow immediately from Lemma B.2. Indeed, for the case of pure-DP ($\delta = 0$), the choice of hyper-parameters in Algorithm 2 and the guarantees of Algorithm 1 (Proposition 1) imply that $\rho \leq L \cdot O(\frac{\sqrt{\log(1/\beta)}\log n_0}{\sqrt{n_0}} + \frac{Td\log(1/\beta)}{n_0\varepsilon})$, which proves Theorem 2. Similarly, Theorem 3 follows by using the guarantees of of Algorithm 1 for approximate $(\varepsilon,\delta)$-DP, that ism Proposition 2, which gives $\rho \leq L \cdot O\left( \frac{\sqrt{\log(1/\beta)}\log n_0}{\sqrt{n_0}} + \frac{T\sqrt{d\log(1/\delta)\log(1/\beta)}}{n_0\varepsilon} \right)$.

Note that our choice of stepsize at each iterate implies that Theorem 2 guarantees the desired utility with probability at least $1 - \beta^2$, hence the final utility guarantee holds with probability at least $1 - T\beta^2 \geq 1 - \beta$.

It remains to verify $T \geq i_0 + 1$. Note that by choosing $T \geq \frac{2\log(D_0^{\kappa-1}\lambda/\rho)}{\kappa-1}$, we get that $D_T \leq (\frac{\kappa 2^\kappa \rho}{\lambda})^{\frac{1}{\kappa-1}}$, hence $T \geq i_0 + 1$. As we have $\rho \geq L/\sqrt{n_0}$ (non-private error) and $D_0^{\kappa-1} \leq L/\lambda$ in our setting, we get that choosing $T = \frac{2\log n}{\kappa-1}$ gives the claim.

## C  Proofs of Section 5

In this section, we provide the proofs for our lower bound under privacy constraints for functions with growth. This section is organized as follows: we prove in Appendix C.1, the lower bounds under pure-DP and in Appendix C.2, the lower bounds under approximate-DP. Within Appendix C.1, we distinguish between $\kappa \geq 2$ (Appendix C.1.1) and $\kappa \in (1,2)$ (Appendix C.1.2).

### C.1  Proofs of Section 5.1

#### C.1.1  Proof of Theorem 4

As we preview in the main text, the proof combines the (non-private) information-theoretic lower bounds of Theorem 8 with the (private) lower bound on ERM of Theorem 9. Finally, we show in Proposition 4 that privately solving SCO is harder than privately solving ERM, concluding the proof of the theorem. We restate the theorem and prove these results in sequence.

**Theorem 4** (Lower bound for $\varepsilon$-DP, $\kappa \geq 2$). *Let $d \geq 1$, $\mathcal{X} = \mathbb{B}_2^d(R)$, $\mathbb{S} = \{\pm e_j\}_{j \leq d}$, $\kappa \geq 2$ and $n \in \mathbb{N}$. Let $\mathcal{P}$ be the set of distributions on $\mathbb{S}$. Assume that*

$$2^{\kappa-1} \leq \frac{L}{\lambda}\frac{1}{R^{\kappa-1}} \leq 2^{\kappa-1}\sqrt{96n} \text{ and } n\varepsilon \geq \frac{1}{\sqrt{3}}$$

*The following lower bound holds*

$$\mathfrak{M}_n(\mathcal{X}, \mathcal{P}, \mathcal{F}^\kappa, \epsilon) \geq \frac{1}{\lambda^{\frac{1}{\kappa-1}}} \tilde{\Omega}\left( \left(\frac{L}{\sqrt{n}}\right)^{\frac{\kappa}{(\kappa-1)}} + \left(\frac{Ld}{n\varepsilon}\right)^{\frac{\kappa}{\kappa-1}} \right). \tag{7}$$

**Non-private lower bound** We begin the proof of Theorem 4 by proving a (non-private) information-theoretic lower bound for minimizing functions with $\kappa \geq 2$-growth. We use the standard reduction from estimation to testing [see 33, Appendix A.1] in conjunction with Fano's method [42, 44].

**Theorem 8** (Non-private lower bound). *Let $d \geq 1$, $\mathcal{X} = \mathbb{B}_2^d(R)$, $\mathbb{S} = \{\pm e_j\}_{j\leq d}$, $\kappa \geq 2$ and $n \in \mathbb{N}$. Let $\mathcal{P}$ be the set of distributions on $\mathbb{S}$. Assume that*

$$2^{\kappa-1} \leq \frac{L}{\lambda}\frac{1}{R^{\kappa-1}} \leq 2^{\kappa-1}\sqrt{96n}.$$

*The following lower bound holds*

$$\mathfrak{M}_n(\mathcal{X}, \mathcal{P}, \mathcal{F}^\kappa) \gtrsim \frac{1}{\lambda^{\frac{1}{\kappa-1}}} \left(\frac{L}{\sqrt{n}}\right)^{\frac{\kappa}{(\kappa-1)}}.$$

*Proof.* For $\mathcal{V} \subset \{\pm 1\}^d$ let us consider the following function and distribution

$$F(x;s) := \frac{\lambda 2^{\kappa-2}}{\kappa}\|x\|_2^\kappa + \frac{L}{2}\langle x, s\rangle \quad \text{and} \quad X \sim P_v \quad \text{implies} \quad X_j = \begin{cases} v_j e_j & \text{w.p. } \frac{1+\delta}{2} \\ -v_j e_j & \text{w.p. } \frac{1-\delta}{2}. \end{cases}$$

Since the linear term does not affect uniform convexity, Lemma 4 in [39] guarantees that $f_v$ is $(\lambda, \kappa)$-uniformly convex. Furthermore, for $s \in \mathbb{S}$

$$\|\nabla F(x;s)\|_2 \leq \lambda 2^{\kappa-2}R^{\kappa-1} + \frac{L}{2} \leq L,$$

by assumption, so the functions are $L$-Lipschitz and satisfy Assumption 1.

Computing the separation. As $\mathbb{E}_{P_v} S = \frac{\delta}{d}v$, we have

$$f_v(x) = \frac{\lambda 2^{\kappa-2}}{\kappa}\|x\|_2^\kappa + \frac{L\delta}{2d}\langle x, v\rangle.$$

Note that for $u \in \mathbb{R}^d, \sigma > 0$, it holds that

$$\inf_{x\in\mathbb{R}^d} \sigma\frac{\|x\|_2^\kappa}{\kappa} + \langle x, u\rangle = -\frac{1}{\kappa^\star}\left(\frac{1}{\sigma}\right)^{\frac{1}{\kappa-1}}\|u\|^{\frac{\kappa}{\kappa-1}} \quad \text{at} \quad x_u^\star = -\left(\frac{1}{\sigma}\right)^{\frac{1}{\kappa-1}}\left(\frac{1}{\|u\|_2}\right)^{\frac{\kappa-2}{\kappa-1}}u.$$

To make sure that $x_u^\star \in \mathbb{B}_2^d(R)$, we require $\|u\|_2 \leq \sigma R^{\kappa-1}$. After choosing $\delta$, we will see that this holds under the assumptions of the theorem. Let us consider the Gilbert-Varshimov packing of the hypercube: there exists $\mathcal{V} \subset \{\pm 1\}^d$ such that $|\mathcal{V}| = \exp(d/8)$ and $d_{\text{Ham}}(v, v') \geq d/4$ for all $v \neq v' \in \mathcal{V}$. Let us compute the separation

$$\inf_{x\in\mathbb{B}_2^d(R)} \frac{f_v(x) + f_{v'}(x)}{2} = -\frac{1}{4\kappa^\star\lambda^{\frac{1}{\kappa-1}}}\left(\frac{L\delta}{d}\right)^{\frac{\kappa}{\kappa-1}}\left\|\frac{v + v'}{2}\right\|_2^{\frac{\kappa}{\kappa-1}}$$

Note that $\|(v + v')/2\|_2 = \sqrt{d - d_{\text{Ham}}(v, v')} \leq \sqrt{3d/4}$. This yields a separation

$$d_{\text{opt}}(v, v', \mathcal{X}) \geq \frac{1 - (3/4)^{\kappa/(2\kappa-2)}}{2\kappa^\star\lambda^{\frac{1}{\kappa-1}}}\left(\frac{L\delta}{\sqrt{d}}\right)^{\frac{\kappa}{\kappa-1}}.$$

Lower bounding the testing error. In the case of a multiple hypothesis test, we use Fano's method and for $V \sim \mathsf{Uni}\{\mathcal{V}\}$ and $S_1^n | V = v \overset{\text{iid}}{\sim} P_v$, Fano's inequality guarantees

$$\inf_{\psi:\mathbb{S}^n\to\mathcal{V}} \Pr(\psi(S_1^n) \neq V) \geq 1 - \frac{I(S_1^n; V) + \ln 2}{\ln|\mathcal{V}|},$$

where $\mathsf{I}(X;Y)$ is the Shannon mutual information between $X$ and $Y$. In our case, we have $\ln|\mathcal{V}| \geq d/8$ and $\mathsf{I}(S_1^n;V) \leq n\max_{v \neq v'} D_{\mathrm{kl}}(P_v\|P_{v'}) \leq 3n\delta^2$. In the case $d \geq 48\ln 2$, we choose $\delta = \sqrt{d/(24n)}$. We handle the one-dimensional case thereafter. For this $\delta$, we have

$$\mathfrak{M}_n(\mathcal{X},\mathcal{P},\mathcal{F}^\kappa) \geq \frac{1 - \left(\frac{3}{4}\right)^{\frac{\kappa}{2\kappa-2}}}{4\kappa^\star(24)^{\frac{\kappa}{2\kappa-2}}} \frac{1}{\lambda^{\frac{1}{\kappa-1}}} \left(\frac{L^2}{n}\right)^{\frac{\kappa}{2\kappa-2}}.$$

For this choice of $\delta$, the assumption on $n$ ensures that the minimum remains in $\mathbb{B}_2^d(R)$.

One-dimensional lower bound with Le Cam's method. Since Fano's method requires $d \geq 48\ln 2$, we finish the proof by providing a lower bound for $d = 1$ using Le Cam's method. We use the same family of functions in one dimension, i.e. $\mathbb{S} = \{\pm 1\}$, $v \in \{\pm 1\}$ and for $\delta \in [0,1]$ define

$$F(x;s) = \frac{\lambda 2^{\kappa-2}}{\kappa}|x|^\kappa + \frac{L}{2}s \cdot x \ \text{ and } \ X \sim P_v \ \text{ implies } \ X = \begin{cases} v & \text{w.p. } \frac{1+\delta}{2} \\ -v & \text{w.p. } \frac{1-\delta}{2}. \end{cases}$$

As this is the one-dimensional analog of the previous construction, $F$ remains $L$-lipschitz and $f$ has $(\lambda,\kappa)$-growth. A calculation yields that the separation is

$$d_{\mathsf{opt}}(1,-1,\mathcal{X}) \geq \frac{1}{2\lambda^{\frac{1}{\kappa-1}}}(L\delta)^{\frac{\kappa}{\kappa-1}},$$

where we used that $\kappa^\star \in [1,2]$. For $V \sim \mathsf{Uni}\{-1,1\}$ and $S_1^n|V = v \overset{\text{iid}}{\sim} P_v$. Le Cam's lemma in conjunction with Pinsker's inequality yields that

$$\inf_{\psi:\mathbb{S}^n \to \{-1,1\}} \Pr(\psi(S_1^n) \neq V) = \frac{1}{2}(1 - \|P_1^n - P_{-1}^n\|_{\mathrm{TV}}) \geq \frac{1}{2}(1 - \sqrt{\tfrac{n}{2}D_{\mathrm{kl}}(P_1\|P_{-1})}).$$

In our case, we have $D_{\mathrm{kl}}(P_1\|P_{-1}) = \delta\ln\frac{1+\delta}{1-\delta} \leq 3\delta^2$ for $\delta \in [0,1/2]$. We set $\delta = 1/\sqrt{6n}$, which yields the final result in one dimension

$$\mathfrak{M}_n([-1,1],\mathcal{P},\mathcal{F}_{d=1}^\kappa) = \Omega\left(\frac{1}{\lambda^{\frac{1}{\kappa-1}}}\left(\frac{L}{\sqrt{n}}\right)^{\frac{\kappa}{\kappa-1}}.\right)$$

$\square$

**Privatizing the lower bound via a packing argument** We now show how this construction yields a private lower bound via a packing argument. For $d \geq 1$, considering the ERM problem, the following private lower bound holds.

**Theorem 9** (Private lower bound for ERM). *Let $d \geq 1$, $\mathcal{X} = \mathbb{B}_2^d(R)$, $\mathbb{S} = \{\pm e_j\}_{j \leq d}$, $\kappa \geq 2$ and $n \in \mathbb{N}$. Let $\mathcal{P}$ be the set of distributions on $\mathbb{S}$. Assume that*

$$2^{\kappa-1} \leq \frac{L}{\lambda}\frac{1}{R^{\kappa-1}} \leq 2^{\kappa-1}\sqrt{96n}.$$

*Then any $\varepsilon$-DP algorithm $\mathsf{A}$ has*

$$\sup_{\mathcal{S} \in \mathbb{S}^n} \mathbb{E}\left[f_{\mathcal{S}}(\mathsf{A}(\mathcal{S})) - \min_{x \in \mathcal{X}} f_{\mathcal{S}}(x)\right] \gtrsim \frac{1}{\lambda^{\frac{1}{\kappa-1}}}\left(\frac{Ld}{n\varepsilon}\right)^{\frac{\kappa}{\kappa-1}}.$$

*Proof.* First, note that it is enough to prove the following lower bound

$$\sup_{\mathcal{S} \in \mathbb{S}^n} \mathbb{E}\left[\|\mathsf{A}(\mathcal{S}) - x^\star\|_2\right] \gtrsim \frac{1}{\lambda^{\frac{1}{\kappa-1}}}\left(\frac{Ld}{n\varepsilon}\right)^{\frac{1}{\kappa-1}}. \tag{9}$$

Indeed, this implies that

$$\sup_{\mathcal{S} \in \mathbb{S}^n} \mathbb{E}\left[f(\mathsf{A}(\mathcal{S})) - \min_{x \in \mathcal{X}} f(x)\right] \geq \frac{\lambda}{\kappa}\sup_{\mathcal{S} \in \mathbb{S}^n} \mathbb{E}\left[\|\mathsf{A}(\mathcal{S}) - x^\star\|_2^\kappa\right]$$

$$\gtrsim \frac{1}{\kappa\lambda^{\frac{1}{\kappa-1}}}\left(\frac{Ld}{n\varepsilon}\right)^{\frac{\kappa}{\kappa-1}}.$$

Let us now prove the lower bound (9). To this end, we consider the function $F(x;s) := \frac{\lambda 2^{\kappa-2}}{\kappa}\|x\|_2^\kappa + \frac{L}{2}\langle x,s\rangle$ where $\|s\|_2 \le 1$. We now construct $M$ datasets $\mathcal{S}_1,\dots,\mathcal{S}_M$ as follows. Let $v_1,\dots,v_M \in \left\{\pm\frac{1}{\sqrt{d}}\right\}^d$ be the Gilbert-Varshimov packing of the hypercube: that is, $M \ge \exp(d/8)$ and $d_{\mathrm{Ham}}(v_i,v_j) \ge d/4$ for all $i \ne j$. We define $\mathcal{S}_i = (\underbrace{v_i,\dots,v_i}_{d/20\varepsilon},0,\dots,0)$. Note that

$d_{\mathrm{Ham}}(S_i,S_j) \le d/20\varepsilon$ and that $f(x;\mathcal{S}_i) = \frac{\lambda 2^{\kappa-2}}{\kappa}\|x\|_2^\kappa + \frac{L}{2}\frac{d}{20n\varepsilon}\langle x,v_i\rangle$, hence

$$x_i^\star = -\left(\frac{1}{\lambda 2^{\kappa-2}}\right)^{\frac{1}{\kappa-1}}\left(\frac{40n\varepsilon}{Ld}\right)^{\frac{\kappa-2}{\kappa-1}}\frac{Ld}{40n\varepsilon}v_i.$$

Therefore we have

$$\|x_i^\star - x_j^\star\|_2^2 \ge \left(\frac{1}{\lambda 2^{\kappa-2}}\right)^{\frac{2}{\kappa-1}}\left(\frac{40n\varepsilon}{Ld}\right)^{\frac{2(\kappa-2)}{\kappa-1}}\frac{L^2 d^2}{1600 n^2 \varepsilon^2}$$

$$\gtrsim \left(\frac{1}{\lambda 2^{\kappa-2}}\right)^{\frac{2}{\kappa-1}}\left(\frac{Ld}{n\varepsilon}\right)^{\frac{2}{\kappa-1}} := \rho^2.$$

We are now ready to finish the proof using packing-based arguments [27]. Assume by contradiction there is an $\varepsilon$-DP algorithm A such that

$$\sup_{1\le i\le M}\mathbb{E}\left[\|\mathsf{A}(\mathcal{S}_i) - x_i^\star\|_2\right] \le \rho/20.$$

Let $B_i = \{x \in \mathcal{X} : \|x - x_i^\star\|_2 \le \rho/2\}$. Note that the sets $B_i$ are disjoint and that Markov inequality implies

$$\Pr(\mathsf{A}(\mathcal{S}_i) \in B_i) = \Pr(\|\mathsf{A}(\mathcal{S}_i) - x_i^\star\|_2 \le \rho/2) \ge 9/10.$$

Thus, the privacy constraint now gives

$$1 \ge \sum_{i=1}^M \Pr(\mathsf{A}(x_1) \in B_i)$$

$$\ge \Pr(\mathsf{A}(x_1) \in B_1) + e^{-d/20}\sum_{i=2}^M \Pr(\mathsf{A}(x_i) \in B_i)$$

$$\ge \frac{9}{10}(1 + e^{-d/20}(M-1)),$$

where the second inequality follows since $d_{\mathrm{Ham}}(\mathcal{S}_i,\mathcal{S}_j) \le d/20\varepsilon$. This gives a contradiction for $d \ge 20$ as $M \ge \exp(d/8)$. For $d = 1$, we can repeat the same arguments with $M = 2$ to get the desired lower bound. $\qquad\square$

**Reduction from $\varepsilon$-DP ERM to $\varepsilon$-DP SCO** We conclude the proof of the theorem by proving that SCO under privacy constraints is strictly harder than ERM. This is similar to Appendix C in [8] but we require it for pure-DP constraints. We make this formal in here.

We have the following lemma.

**Proposition 4.** *Let $0 < \beta \le 1/n$. Assume A is an $\frac{\varepsilon}{2\log(2/\beta)}$-DP algorithm that for a sample $\mathcal{S} = (S_1,\dots,S_n)$ with $S_1^n \overset{\mathrm{iid}}{\sim} P$ achieves with probability $1 - \beta/2$ error*

$$f(\mathsf{A}(\mathcal{S})) - \min_{x\in\mathcal{X}}f(x) \le \gamma.$$

*Then there is an $\varepsilon$-DP algorithm $\mathsf{A}'$ such that for any dataset $\mathcal{S} \in \mathbb{S}^n$ has with probability $1 - \beta$,*

$$f_\mathcal{S}(\mathsf{A}'(\mathcal{S})) - \min_{x\in\mathcal{X}}f_\mathcal{S}(x) \le \gamma.$$

*Proof.* Given the algorithm A, we define $\mathsf{A}'$ as follows. For an input $\mathcal{S} \in \mathbb{S}^n$, let $P_\mathcal{S}$ be the empirical distribution of $\mathcal{S}$. Then, $\mathsf{A}'$ proceeds as follows:

1. Sample a new dataset $\mathcal{S}_1 = (S'_1, \dots, S'_n)$ where $S'_i \sim P_{\mathcal{S}}$

2. If there is a sample $S_i$ that was sampled more than $k = 2\log(2/\beta)$ times, return $0$

3. Else, return $\mathsf{A}(\mathcal{S}_1)$

We need to prove that $\mathsf{A}'$ is $\varepsilon$-DP and that it has the desired utility. For utility, note that $\mathsf{A}'$ returns $0$ at step 2 with probability at most $\beta/2$, since we have for every $1 \leq i \leq n$

$$\Pr(s_i \text{ used more than } k \text{ times}) = \Pr\left( \sum_{j=1}^n Z_i \geq k \right)$$

$$\leq 2^{-k} \leq \beta^2/2,$$

where $Z_j \sim \mathsf{Bernoulli}(p)$ with $p = 1/n$, and the second inequality follows from Chernoff [36, Thm. 4.4] and $\beta \leq 1/10$. Applying a union bound over all samples, we get that step 2 returns $0$ with probability at most $\beta/2$ as $\beta \leq 1/n$. Moreover, Algorithm $\mathsf{A}$ fails with probability at most $\beta/2$. Therefore, as $f_{\mathcal{S}}(x) = \mathbb{E}_{S \sim P_{\mathcal{S}}}[F(x; S)]$, we have with probability at least $1 - \beta$,

$$f_{\mathcal{S}}(\mathsf{A}'(\mathcal{S})) - \min_{x \in \mathcal{X}} f_{\mathcal{S}}(x) \leq \gamma.$$

Let us now prove privacy. Assume we run algorithm $\mathsf{A}'$ on two neighboring datasets $\mathcal{S}, \mathcal{S}'$, and let $\mathcal{S}_1, \mathcal{S}'_1$ be the datasets produced at step 1. Let $B$ denote the event that there was a sample $s_i$ that was used more than $k$ times (note that this does not depend on the input). Then for any measurable $\mathcal{O}$,

$$\Pr(\mathsf{A}'(\mathcal{S}) \in \mathcal{O}) = \Pr(\mathsf{A}'(\mathcal{S}) \in \mathcal{O} \mid B)\Pr(B) + \Pr(\mathsf{A}'(\mathcal{S}) \in \mathcal{O} \mid B^c)\Pr(B^c)$$

$$\leq e^\varepsilon \Pr(\mathsf{A}'(\mathcal{S}') \in \mathcal{O} \mid B)\Pr(B) + \Pr(\mathsf{A}'(\mathcal{S}') \in \mathcal{O} \mid B^c)\Pr(B^c)$$

$$\leq e^\varepsilon \Pr(\mathsf{A}'(\mathcal{S}') \in \mathcal{O}),$$

where the first inequality follows from group privacy since $d_{\mathrm{Ham}}(\mathcal{S}_1, \mathcal{S}'_1) \leq k$ and $\mathsf{A}$ is $\varepsilon/k$-DP. This completes the proof. $\qquad\square$

### C.1.2 Proof of Theorem 5

**Theorem 5** (Lower bound for $\varepsilon$-DP, $\kappa \in (1, 2]$ )**.** *Let* $d = 1$, $\mathbb{S} = \{-1, +1\}$, $\kappa \in (1, 2]$, $\lambda = 1$, $L = 2$, *and* $n \in \mathbb{N}$. *There exists a collection of distributions* $\mathcal{P}$ *such that, whenever* $n\varepsilon \geq 1/\sqrt{3}$, *it holds that*

$$\mathfrak{M}_n([-1, 1], \mathcal{P}, \mathcal{F}_{d=1}^\kappa, \epsilon) = \Omega\left\{ \left(\frac{1}{\sqrt{n}}\right)^{\frac{\kappa}{\kappa-1}} + \left(\frac{1}{n\varepsilon}\right)^{\frac{\kappa}{\kappa-1}} \right\}. \tag{8}$$

*Proof.* We follow the same reduction that we used in the proof of Theorem 8. For $\delta \in [0, 1/2]$, we again consider $P_v = 1$ with probability $\frac{1+\delta v}{2}$ and $-1$ otherwise. For $a \in [0, 1]$ to be defined later, we construct the following function

$$F(x; +1) = \begin{cases} |x - a| & \text{if } x \leq a \\ |x - a|^\kappa & \text{if } x \geq a \end{cases} \text{ and } F(x; -1) = \begin{cases} |x + a|^\kappa & \text{if } x \leq -a \\ |x + a| & \text{if } x \geq -a \end{cases}$$

Computing the separation. First, let us compute the separation $d_{\mathsf{opt}}(v, v', \mathcal{X})$. We will then choose $a$ to ensure $f_v$ has $\kappa$-growth. By symmetry, assume $v = 1$. $f_v$ is increasing on $[a, 1]$ and decreasing on $[-1, -a]$, thus the minimum belongs to $[-a, a]$ and by inspection, is attained at $x = a$ with value $a(1 - \delta)$. Similarly, the minimum of $f_{+1}(x) + f_{-1}(x)$ is attained on $[-a, a]$ with value $2a$. This yields

$$d_{\mathsf{opt}}(v, v', \mathcal{X}) = 2a - 2a(1 - \delta) = 2a\delta.$$

Let us now pick $a$ such that $f_v$ has $\kappa$-growth. Again, by symmetry we only treat the $v = 1$ case. We have

for $x \geq a$, $f_v(x) - f_v^\star = \dfrac{1+\delta}{2}(x-a)^\kappa + \dfrac{1-\delta}{2}(x+a) - a(1-\delta) = \dfrac{1+\delta}{2}(x-a)^\kappa + \dfrac{1-\delta}{2}(x-a) \geq |x-a|^\kappa,$

where the last inequality is because $(x - a) \leq 1$ and so $(x - a) \geq (x - a)^\kappa$ for $\kappa > 1$. In the second case, we have

$$\text{for } x \in [-a, a], f_v(x) - f_v^\star = \delta(a - x).$$

It holds that $\delta(a - x) \geq (a - x)^\kappa$ for all $x \in [-a, a]$ iff $a \leq \frac{1}{2} \delta^{\frac{1}{\kappa - 1}}$. As a result, we set $a = \frac{1}{2} \delta^{\frac{1}{\kappa - 1}}$. Finally, for $x \in [-1, -a]$, we define

$$h(x) := \frac{1 + \delta}{2} |x - a| + \frac{1 - \delta}{2} |x + a|^\kappa - a(1 - \delta) - \frac{1}{\kappa} |x - a|^\kappa \text{ for } x \in [-1, -a].$$

We wish to prove that $h(x) \geq 0$. First of, note that $h(-a) = \delta^{\frac{\kappa}{\kappa - 1}} \left( \frac{1}{2} + \frac{1}{2} - \frac{1}{\kappa} \right) > 0$, whenever $\kappa > 1$. Let us show that $h(x)$ is decreasing on $[-1, -a]$ which suffices to conclude the proof. We have

$$h'(x) = -\frac{1 + \delta}{2} - \frac{\kappa(1 - \delta)}{2} |x + a|^{\kappa - 1} + |x - a|^{\kappa - 1}.$$

First of, note that $h'(-a) = -\frac{1+\delta}{2} + \delta \leq 0$ and $h'(-1) < 0$, thus it suffices to show that if $h'$ has an extremum then is it negative. An extremum of this function is a point $x^\star$ such that

$$|a - x^\star| = \left( \frac{\kappa(1 - \delta)}{2} \right)^{\frac{1}{\kappa - 2}} |a + x^\star|,$$

which yields that

$$h'(x^\star) = |a + x^\star|^{\kappa - 1} \left( \frac{\kappa(1 - \delta)}{2} \right) \left[ \left( \frac{\kappa(1 - \delta)}{2} \right)^{\frac{1}{\kappa - 2}} - 1 \right] - \frac{1 + \delta}{2} \leq 0,$$

as $\kappa \leq 2$. This calculation shows that $f_v$ has $(1, \kappa)$-growth. Finally note that the function is $\kappa \leq 2$-Lipschitz as desired.

Lower bounding the testing error. It remains to choose the value of $\delta$. Since we require a lower bound under privacy constraints, in contrast to the one-dimensional section of the proof of Theorem 8, we require the following privatized version of Le Cam's lemma from [6]

**Proposition 5.** *[6, Thm. 2] Let $\mathsf{A} \in \mathcal{A}^\varepsilon$ be an $\epsilon$-DP mechanism from $\mathbb{S}^n \to \mathcal{X}$. It holds that*

$$\inf_{\psi : \mathcal{X} \to \{-1, 1\}} \inf_{\mathsf{A} \in \mathcal{A}^\epsilon} \Pr(\psi(\mathsf{A}(S_1^n)) \neq V) \geq \frac{1}{2} \left( 1 - \min\{ 2n\varepsilon \| P_1 - P_{-1} \|_{\mathrm{TV}}, \| P_{-1}^n - P_1^n \|_{\mathrm{TV}} \} \right).$$

With this result, we set $\delta = \max\{1/\sqrt{6n}, 1/(2\sqrt{3}n\varepsilon)\}$ and lower bound $\max\{a, b\}$ by $a + b$ for readability, which concludes the proof of the theorem. □

## C.2 Proof for Section 5.2

**Proposition 3** (Solving ERM with $\kappa$-growth implies solving any convex ERM). *Let $\kappa \geq 2$. Assume there exists an $(\epsilon, \delta)$ mechanism $\mathsf{A}$ such that for any $L$-Lipschitz loss $G$ on $\mathcal{Y}$ and dataset $\mathcal{S}$ such that $g_\mathcal{S}(x) := \frac{1}{n} \sum_{s \in \mathcal{S}} G(x; s)$ exhibits $(\lambda, \kappa)$-growth, the mechanism achieves excess loss*

$$\mathbb{E}[g_\mathcal{S}(\mathsf{A}(\mathcal{S}, G, \mathcal{Y}))] - \inf_{y' \in \mathcal{Y}} g_\mathcal{S}(y') \leq \frac{1}{\lambda^{\frac{1}{\kappa - 1}}} \Delta(n, L, \epsilon, \delta).$$

*Then, we can construct an $(\varepsilon, \delta)$-DP mechanism $\mathsf{A}'$ such that for any $L$-Lipschitz loss $f$, the mechanism achieves excess loss*

$$\mathbb{E}[f_\mathcal{S}(\mathsf{A}'(\mathcal{S}))] - \inf_{x' \in \mathcal{X}} f_\mathcal{S}(x') \leq O\left( D[\Delta(n, L, \epsilon/k, \delta/k)]^{\frac{\kappa - 1}{\kappa}} \right),$$

*where $k$ is the smallest integer such that $k \geq \log \left[ \frac{\kappa^{\frac{1}{\kappa - 1}} L^{\frac{\kappa}{\kappa - 1}}}{2^{2\kappa - 3} \Delta(n, L, \varepsilon/k, \delta/k)} \right]$.*

*Proof of Proposition 3.* Let us first show how to construct the mechanism $\mathsf{A}'$. Let $k \in \mathbb{N}$ be such that $k \geq \log\left\lceil \frac{\kappa^{\frac{1}{\kappa-1}} L^{\frac{\kappa}{\kappa-1}}}{2^{2\kappa-3}\Delta(n,L,\varepsilon/k,\delta)} \right\rceil$ and let $\{\lambda_i\}_{i \in [k]}$ be a collection of positive scalars. Set $x_0 \in \mathcal{X}$, for $i \in \{1, \ldots, k\}$

define $G_i(x;s) = F(x;s) + \frac{\lambda_i \cdot 2^{\kappa-2}}{\kappa}\|x - x_{i-1}\|_2^\kappa, \mathcal{Y}_i := \left\{ x \in \mathcal{X} : \|x - x_{i-1}\|_2 \leq \left(\frac{L\kappa}{\lambda_i 2^{\kappa-2}}\right)^{\frac{1}{\kappa-1}} \right\}$

and set $x_i = \mathsf{A}(\mathcal{S}, G_i, \mathcal{Y}_i)$, with privacy $(\varepsilon/k, \delta/k)$.

Finally, define $\mathsf{A}'(\mathcal{S}) = x_k$. Standard composition theorems [20] guarantee that $\mathsf{A}'$ is $(\varepsilon, \delta)$-DP. Let us analyze its utility; we drop the dependence of $\Delta$ on other variables when clear from context. First of, since $\kappa$ is a constant, note that $G_i$ is $c_0 L$-Lipschitz with $c_0 < \infty$ a numerical constant. For simplicity, we define $g_i(x) := \frac{1}{n}\sum_{s \in \mathcal{S}} G_i(x;s)$ and $x_i^\star = \operatorname{argmin}_{x \in \mathcal{Y}_i} g_i(x)$. It holds that $g_i$ is $(\lambda_i 2^{\kappa-2}, \kappa)$-uniformly-convex and thus the following growth condition holds

$$\frac{\lambda_i}{\kappa}\mathbb{E}\|x_i - x_i^\star\|_2^\kappa \leq \mathbb{E}[g_i(x_i)] - g_i(x_i^\star) \leq \frac{1}{\lambda_i^{\frac{1}{\kappa-1}}}\Delta.$$

Also note that for any point $y \in \mathcal{Y}_i$, it holds that

$$f_\mathcal{S}(x_i^\star) - f(y) \leq \frac{\lambda_i 2^{\kappa-2}}{\kappa}\|x_{i-1} - y\|_2^\kappa.$$

Finally, let us bound the distance to the optimum of $f_\mathcal{S}$ at the final iterate. We have

$$\frac{\lambda_k}{\kappa}\|x_k - x_k^\star\|_2^\kappa \leq g_k(x_k) - g_k(x_k^\star) \leq c_0 L\|x_k - x_k^\star\|_2 \text{ which yields } \|x_k - x_k^\star\|_2 \leq \left(\frac{c_0 L\kappa}{\lambda_k}\right)^{\frac{1}{\kappa-1}}.$$

Let us put the pieces together: for $\lambda > 0$ to be determined later and $\nu = \kappa - 1$, set $\lambda_i = 2^{-\nu i}\lambda$. After $k$ rounds and denoting $x_0^\star = \inf_{x \in \mathcal{X}} f_\mathcal{S}(x)$, we have

$$\mathbb{E}[f_\mathcal{S}(x_k)] - f_\mathcal{S}(x^\star) = \sum_{i=1}^k \mathbb{E}\big[f_\mathcal{S}(x_i^\star) - f_\mathcal{S}(x_{i-1}^\star)\big] + \mathbb{E}[f_\mathcal{S}(x_k) - f_\mathcal{S}(x_k^\star)]$$

$$\leq \sum_{i=1}^k \frac{\lambda_i 2^{\kappa-2}}{\kappa}\mathbb{E}\|x_{i-1} - x_{i-1}^\star\|_2^\kappa + L\left(\frac{c_0 L\kappa}{\lambda_k}\right)^{\frac{1}{\kappa-1}}$$

$$\leq \frac{\lambda D^\kappa}{\kappa} + \sum_{i=2}^k \frac{\lambda_i 2^{\kappa-2}}{\lambda_{i-1}^{\frac{\kappa}{\kappa-1}}}\Delta + L\left(\frac{c_0 L\kappa}{\lambda_k}\right)^{\frac{1}{\kappa-1}}$$

$$= \frac{\lambda D^\kappa}{\kappa} + \frac{\Delta 2^{\kappa-2}}{\lambda^{\frac{1}{\kappa-1}}}\sum_{i=2}^k 2^{-\frac{\nu}{\kappa-1}(i-\kappa)} + L\left(\frac{c_0 L\kappa}{\lambda}\right)^{\frac{1}{\kappa-1}}2^{-\frac{\nu}{\kappa-1}k}$$

$$\leq \frac{\lambda D^\kappa}{\kappa} + 2^{2\kappa-3}\frac{\Delta}{\lambda^{\frac{1}{\kappa-1}}} + \frac{\kappa^{\frac{1}{\kappa-1}}(c_0 L)^{\frac{\kappa}{\kappa-1}}2^{-k}}{\lambda^{\frac{1}{\kappa-1}}}.$$

Finally, note that

$$k \geq \left\lceil \log\left[\frac{\kappa^{\frac{1}{\kappa-1}}(c_0 L)^{\frac{\kappa}{\kappa-1}}}{2^{2\kappa-3}\Delta}\right]\right\rceil \text{ so that } \frac{\kappa^{\frac{1}{\kappa-1}}(c_0 L)^{\frac{\kappa}{\kappa-1}}2^{-k}}{\lambda^{\frac{1}{\kappa-1}}} \leq 2^{2\kappa-3}\frac{\Delta}{\lambda^{\frac{1}{\kappa-1}}}.$$

It then holds that

$$\mathbb{E}[f_\mathcal{S}(x_k)] - f_\mathcal{S}(x^\star) \leq \lambda\frac{D^\kappa}{\kappa} + 4^{\kappa-1}\Delta\frac{1}{\lambda^{\frac{1}{\kappa-1}}}.$$

It remains to pick $\lambda$ to minimize the upper bound above. A calculation yields that for $a, b \geq 0$

$$\inf_{\nu \geq 0} a\nu + \frac{b}{\nu^{\frac{1}{\kappa-1}}} = (\kappa-1)^{1/\kappa} a^{1/\kappa} b^{(\kappa-1)/\kappa} \left[\kappa - 1 + \frac{1}{\kappa-1}\right] \text{ at } \nu^\star = \left(\frac{b}{a(\kappa-1)}\right)^{\frac{\kappa-1}{\kappa}}.$$

Setting $\lambda = 4^{\frac{(\kappa-1)^2}{\kappa}} \left(\frac{\Delta\kappa}{D^\kappa(\kappa-1)}\right)^{(\kappa-1)/\kappa}$ yields the regret bound

$$\mathbb{E}[f_\mathcal{S}(x_k)] - f_\mathcal{S}(x^\star) \leq O(1) D \Delta^{\frac{\kappa-1}{\kappa}}.$$

$\square$

*Proof.* Consider the reduction of Proposition 3. For $c_1 < \infty$ to be determined later, assume by contradiction that there exists an $(\varepsilon, \delta)$ mechanism such that

$$\Delta(n, L, \varepsilon, \delta) \leq c_1 \left(\frac{L\sqrt{d}}{n\varepsilon}\right)^{\frac{\kappa}{\kappa-1}}.$$

Setting $k = \lceil 4\log(n\varepsilon/\sqrt{d})\log\log((n\varepsilon/\sqrt{d})^{\kappa/(\kappa-1)})\rceil$, the condition holds and the result of Proposition 3 guarantees that there exists a numerical constant $c_2 < \infty$ and a mechanism $\mathsf{A}'$ such that

$$\mathbb{E}[f_\mathcal{S}(\mathsf{A}'(\mathcal{S}))] - \inf_{x' \in \mathcal{X}} f_\mathcal{S}(x') \leq c_2 c_1^{\frac{\kappa-1}{\kappa}} kD \frac{L\sqrt{d}}{n\varepsilon}.$$

However, Theorem 5.3 in [7] guarantees that there exists $c_3 > 0$ such that for any $(\varepsilon, \delta)$-DP mechanism $\mathsf{A}''$, it must hold

$$c_3 L D \frac{\sqrt{d}}{n\varepsilon} \leq \mathbb{E}[f_\mathcal{S}(\mathsf{A}''(\mathcal{S}))] - f_\mathcal{S}(x^\star).$$

Setting $c_1 = \frac{1}{2}\left(\frac{c_3}{kc_2}\right)^{\frac{\kappa}{\kappa-1}}$ yields a contradiction and the desired lower bound by noting that $k$ consists only of log factors. $\square$