# OpenReview forum: "Adapting to function difficulty and growth conditions in private optimization"
_NeurIPS.cc/2021/Conference — NeurIPS 2021 Poster_

### Official Review · Reviewer_GACB · 2021-07-14

**Rating:** 7
**Confidence:** 2

**Summary:**

This work proposes private stochastic convex optimization algorithms that adapt to the growth condition of the objective function. This work also establishes several lower bounds that justify the optimality of the proposed algorithms.

**Limitations And Societal Impact:**

The authors have adequately addressed the limitations and potential negative societal impact of their work.

**Main Review:**

This paper is well-written and easy-to-follow. The text has a nice flow of ideas. The developed algorithms only require a lower bound on the actual growth constant (probably need to make this requirement clearer in the abstract) and adapt to the true growth condition to improve the privacy rate, which is a significant contribution. The prior work/background is discussed at length throughout the paper and the theorems are attached with proof sketches and insights, which are highly appreciated. I only have some  minor comments as follows.

- I wonder how the growth constant $\kappa$ can be efficiently estimated and how a very rough estimate of $\kappa$ would affect the rate.
- Typos: Line 143, different fonts for A_1 and A_k; In Algorithm 1, at line 10, remove "algorithm"; Line 484, developing.

**Time Spent Reviewing:**

4

---

> ### Author Response · Authors · 2021-08-10
> **Response to review**
>
> We would like to thank the reviewer for their valuable time and feedback and for saying that the paper is “well-written”, “has a nice flow of ideas” and that it constitutes a “significant contribution”.
>
> - **[Question about estimating $\kappa$]**: We do not currently know of ways to estimate the growth constant $\kappa$, which is the main motivation for our adaptive algorithms that obtain the correct rate without knowing its value. Even in standard (non-private) optimization, this desire for robustness and adaptivity is central; recall, e.g., Nemirovski et al.’s “Robust Stochastic Approximation,” where they demonstrate that standard stochastic gradient algorithms may converge exponentially more slowly with a mis-estimated strong convexity constant. In our case, if we underestimate $\kappa$ by $\tilde{\kappa} \le \kappa$, then we need to pay an extra $\frac{\kappa - 1}{\tilde{\kappa} - 1}$ in the convergence rate (inside the power); however if we overestimate $\kappa$, then we achieve the rates corresponding to $\tilde{\kappa}$ which has a worse power $\frac{\tilde{\kappa}}{\tilde{\kappa} - 1}$ instead of $\frac{\kappa}{\kappa-1}$. In practice, setting $\tilde{\kappa}$ to be a constant factor above one should generally be a good choice. We will make this clearer in the next version of the paper.
> - **[Typos]**: thank you for pointing these out, we will fix this in the next version of the paper.

---

### Official Review · Reviewer_aQoc · 2021-07-19

**Rating:** 6
**Confidence:** 3

**Summary:**

This paper presents a differentially private mechanism for releasing data and an optimisation algorithm that can use the released data and minimise a convex function. While there are many such algorithms in prior work, the current manuscript provides an algorithm that is adaptive and has convergence rate that improves with the "growth" of the function around the optimum point. Furthermore, the authors present matching lower bounds, which show that their algorithm is close to optimal.


**Limitations And Societal Impact:**

Sufficiently addressed.

**Main Review:**

*Contribution*
The main contribution of the paper is two-fold: an upper bound which attains the claimed convergence rate and the lower bound. I was unable to read the details about the lower bound in the supplementary material and the description in the main text didn't provide a clear picture to me. So I will focus on the upper bound. The algorithm is simple to describe: In each iteration, the algorithm limits the learning rate, reduces the domain to the neighborhood of the previous point, and repeats multiple iterations of a local optimization procedure. The local optimization procedure uses ERM to minimise an $\ell_2$-regularized version of the function with the regularizer weight updated exponentially in each inner iteration. Just before releasing the point, a Laplace noise is added to the output, which ensures differential privacy. I found it remarkable that the overall algorithm is so simple and can be (perhaps) implemented even in practice.

I was able to roughly verify the proof, but I couldn't find a crisp high-level description of the algorithm. For instance, what was the role of exponentially decreasing variance in step 9? What is the tradeoff between privacy and convergence rate as a function of the noise parameter chosen in this step?


*Originality*
I found the problem and the result very interesting. The algorithm proposed (algorithms 1,2) are very simple, and somewhat intuitively clear. However, owing to my lack of knowledge about the prior work and lack of time to dig deeper, I was unable to evaluate the contribution in comparison with prior work. Nonetheless, the authors have provided adequate references to prior work, and I think the work is sufficiently novel.

*Quality and clarity*
The paper is well-written, especially Section 4, and presents results on an important problem. I was a bit lost in Section 3 -- even the statement of Theorem was not very clear since the algorithm $A_{\tt gr-inv}(S)$ was not very clear. But overall the work is of high quality.


*Significance*
There is broad interest in studying optimization under privacy and similar constraints, and this paper presents a new algorithm for that setting. Even if one ignores the theoretical optimality, the paper provides an interesting algorithm for a practitioner to build on.



**Time Spent Reviewing:**

3

---

> ### Author Response · Authors · 2021-08-10
> **Response to review**
>
> We would like to thank the reviewer for their valuable time and feedback and for finding the “the problem and the result very interesting”, that “the work is of high-quality” and that the “algorithms proposed are very simple and intuitively clear”.
>
> - **[Role of decreasing variance]**: to guarantee differential privacy, we need to add noise commensurate to the sensitivity. In the case of optimization with privacy, the challenge is to add the right amount of noise without compromising utility. As the radius shrinks exponentially at each step, the required amount of noise shrinks accordingly, which explains the exponentially decreasing variance. We will make this intuition clearer in the next version of the paper.
> - **[Trade-off between privacy and convergence rate]**: $\varepsilon$ controls the privacy level of our algorithm. At each step, the variance is proportional to $\frac{1}{\varepsilon}$ and the amount of noise controls how private our algorithm is. Choosing smaller $\varepsilon$ (or equivalently larger variance) will result in larger error as made explicit by our utility analysis of Proposition 2.
> - **[Clarification on Section 3]**: The (approximate) inverse sensitivity mechanism essentially (privately) finds an approximate minimizer of $||\nabla f_S(x) ||_2$ using the standard exponential mechanism. Under growth conditions, this yields the convergence rate we present in Theorem 1. We will clarify this in the next version of the paper.

---

### Official Review · Reviewer_pyyH · 2021-07-27

**Rating:** 5
**Confidence:** 5

**Summary:**

This paper studies the SCO with loss function that satisfies growth conditions in the differential privacy model. Firstly, it considers the empirical risk function and provide an algorithm based on the inverse sensitivity mechanism. Then they show an efficient algorithm and the lower bounds of the problem in the both $\epsilon$ and $(\epsilon, \delta)$-DP models.

**Limitations And Societal Impact:**

Limitations:

1) Lower bounds only hold for the case where $k\geq 2$ for general dimension case.
2) As we can see from the result, theoretically, the rate of DP-SCO with growth condition loss is  faster than the general convex loss. However, there is lack of experiments, most of the previous work on SCO with growth condition has experimental studies. So I think there is still improvements.
3)In Algorithm 2, step 10 need to do a projection step onto the set $\mathcal{X}_i$ which is the intersection of the constraint set $\mathcal{X}$ and a ball.  How to get the exact solution of this step. If we can get approximate solution by using such as Dykstra’s algorithm, then how to guarantee the DP and the upper bound? So I don't think it is an efficient algorithm.

Based on the previous concerns, I tend to reject the paper.

**Main Review:**

Advantage:
1)They provide a high probability version of the previous algorithm on DP-SCO with general convex loss functions, which might be used to other problems.
2)They give the first study on DP-SCO with loss functions that satisfy the growth condition and show their upper bounds and lower bounds

**Time Spent Reviewing:**

3hours

---

> ### Author Response · Authors · 2021-08-10
> **Response to review**
>
> We thank the reviewer for their time and valuable feedback.
>
> - **[1. Comment about lower bound]**: First, even in the simpler non-private case, our result for $\kappa<2$ is the first information-theoretic lower bound (this is in contrast to the weaker previous bounds of Ramdas and Singh, which hold only in an oracle model where the optimizer gets stochastic gradients). As we point out in the paper, the lower bound construction we develop for $\kappa \ge 2$ does not (and cannot) exhibit \kappa growth; we therefore develop the novel “two-sided function.” As this construction is more subtle, it appears challenging to extend to the general dimensional case (though of course the claimed lower bound applies in all dimensions).
> - **[2. Comment about experiments]**: Our goal in this investigation is theoretical in nature, aiming to understand tight minimax rates of DP-SCO problems and explore the fundamental limits of adaptivity in private optimization. Certainly deeper experimental work is important--and we plan to do so in the future--though our focus and the constraints of the NeurIPS format necessitate compromises. (And while this may not be a particularly compelling argument, many influential papers in DP-SCO or DP-ERM, e.g. [1,2,3,4] are fully theoretical; this is often the case even in non-private optimization [5,6].)
> - **[3. Comment about inexact projections]**: The only assumption of Algorithm 2 is that we can efficiently solve a strongly-convex ERM problem over a convex constraint set. This is standard in the literature. Solving the strongly-convex ERM approximately (with inexact projections) does not change our results in any meaningful way. To give more details, in the analysis, if we were to do inexact projections (e.g. using Dykstra’s algorithm), it would only increase the sensitivity by the extra error incurred by the inexact projection. As long as this error is on the order of $\frac{4L^2}{\lambda n}$ (that is, the stability of the exact minimizer), this does not (except for numerical constants) change the analysis. Generally, projection onto the intersection of convex sets can be solved efficiently (i.e. with linear rates of convergence; see, for example, the line of work by Bauschke and collaborators on alternating projections), so this does not incur significant additional computational cost. We did not detail this in the submission to keep the focus on the crucial ideas without obfuscation via (to our minds) unnecessary technical details. We will make it clear in the final version with a new section of the Appendix detailing this analysis.
>
> [1] Hilal Asi,, Vitaly Feldman, Tomer Koren, Kunal Talwar, Private Stochastic Convex Optimization: Optimal Rates in  Geometry, ICML 2021.
>
> [2] Raef Bassily, Vitaly Feldman, Kunal Talwar, Abhradeep Thakurta, Private stochastic convex optimization with optimal rates, NeurIPS 2019.
>
> [3] Raef Bassily, Adam Smith, Abhradeep Thakurta. Private empirical risk minimization: Efficient algorithms and tight error bounds, STOC 2014.
>
> [4] Vitaly Feldman, Tomer Koren, Kunal Talwar, Private stochastic convex optimization: optimal rates in linear time, STOC 2020.
>
> [5] A. Ramdas and A. Singh. Optimal rates for stochastic convex optimization under tsybakov noise condition, ICML 2013.
>
> [6] A. Juditsky and Y. Nesterov. Deterministic and stochastic primal-dual subgradient algorithms for uniformly convex minimization, Stochastic Systems 2014.

---

### Decision · Program_Chairs · 2021-09-27

**Decision:**

Accept (Poster)

**Comment:**

Despite some suggestions for additional results that could round out the paper, and presentation suggestions for clarifying the paper's conceptual contributions, I would like to recommend it for acceptance. In particular:

- As an algorithmic paper, providing a new suite of algorithms and complementing them with matching lower bounds tells a complete story. Experiments would be nice, but given the other results I do not believe they are necessary.

- Concerning Reviewer pyyH's comment about inexact projections, the author's proposed modifications should indeed be implemented carefully. Ensuring that the approximate projections do not (even in the worst-case over the projection algorithm) significantly increase sensitivity affects the privacy guarantee of the final algorithm. So this is an important effect to be accounted for in detail.